# Wide-energy programmable microwave plasma-ionization for high-coverage mass spectrometry analysis

Fengjian Chu[1,2], Gaosheng Zhao[3], Wei Wei[2], Nazifi Sani Shuaibu[1], Hongru Feng[2] ✉, Yuanjiang Pan[2] ✉ & Xiaozhi Wang[1] ✉

Although numerous ambient ionization mass spectroscopy technologies have been developed over the past 20 years to address diverse analytical circumstances, a single-ion source technique that can handle all analyte types is still lacking. Here, a wide-energy programmable microwave plasma-ionization mass spectrometry (WPMPI-MS) system is presented, through which MS analysis can achieve high coverage of substances with various characteristics by digitally regulating the microwave energy. In addition, ionization energy can be rapidly scanned using programmable waveforms, enabling the simultaneous detection of biomolecules, heavy metals, non-polar molecules, etc., in seconds. WPMPI-MS performs well in analyzing real samples, rapidly analyzing nine toxicological standards in one drop of serum, and demonstrating good quantification and liquid chromatography coupling capability. The WPMPI-MS has also been used to detect soil extracts, solid pharmaceuticals, and landfill leachate, further demonstrating its robust analytical capabilities for real samples. The prospective uses of the technology in biological and chemical analysis are extensive, and it is anticipated to emerge as a viable alternative to commercially available ion sources.

Ion sources are among the most essential components of mass spectrometry (MS) and have evolved from early electron impact (EI), chemical ionization (CI), and electrospray ionization (ESI) to various types of ambient ionization technologies, providing multidimensional application scenarios for mass spectrometry analysis. As mass spectrometry technology matures and becomes commercial, an increasing number of ionization techniques are being developed in open air due to their unprocessed or minimally pretreatment process, widely suitable to MS instruments, and good analytical performance[1]. In the last 20 years, researchers have introduced and modified dozens of ambient-ionization technologies, which utilize diverse physicochemical processes, such as electrospray[2], laser ablation[3], plasma[4], thermal desorption[5] and vibrational excitation[6], and demonstrated great analytical performance in a variety of

applications, most of which are defined as ambient mass spectrometry (AMS).

Due to the large variances in the chemical characteristics of the analytes, it is difficult to cover all the analytes by a single-ionization technology. For example, ambient-ionization technology based on an electrospray mechanism is usually utilized for nonvolatile polar compounds throughout an extensive mass range[2,7,8], and ionization techniques based on an atmospheric pressure chemical ionization (APCI) mechanism can characterize compounds that are nonpolar or exhibit a low polarity[9-11]. An ion source with full analytical coverage is not only helpful for the analysis of complex samples, leading to the creation of more specific fingerprints, but also eliminates the need for additional mass spectrometry. For example, toxicology centers in advanced hospitals and toxicology laboratories usually contain three mass

[1]College of Information Science and Electronic Engineering, Zhejiang University, Hangzhou 310027, P. R. China. [2]Department of Chemistry, Zhejiang University, Hangzhou 310027, P. R. China. [3]School of Environmental and Chemical Engineering, Shanghai University, Shanghai 200444, P. R. China. ✉e-mail: fenghongru@zju.edu.cn; panyuanjiang@zju.edu.cn; xw224@zju.edu.cn

spectrometers (ESI, EI, and ICP-MS) to address different clinical situations involving poisoned patients. Therefore, the development of advanced ionization techniques to analyze various molecules with higher coverage is a major goal for many analytical chemists.

To address the limited range of ionized molecules and broaden the application of mass spectrometry, researchers have proposed many atmospheric-pressure plasma-ionization techniques. For example, direct analysis in real time (DART)[12], desorption atmospheric-pressure chemical ionization (DAPCI)[13], flowing afterglow atmospheric-pressure glow discharge (FA-APGD)[14], dielectric barrier discharge ionization (DBDI)[15,16], low-temperature plasma (LTP) probe[9], microwave plasma torch (MPT)[17,18], ambient flame ionization (AFI)[19], and ambient electric-arc ionization (AEAI)[20] are widely used for the direct detection of various environmental and biological samples. Based on these rich atmospheric pressure plasma techniques, some researchers have combined different ionization techniques to broaden the analyte molecules that are detectable[21,22]. These composite ion sources utilize distinct and collective ionization modes through parameter adjustments, permitting the identification of substances with diverse characteristics. Even so, these composite ion sources generally only cover polar or nonpolar compounds without elemental analysis capabilities and stepless ionization energy regulation.

In addition, since most ambient ionization technologies utilize soft ionization mechanisms, which typically result in intact molecular ions or a limited number of fragment ions in the mass spectra, researchers are compelled to utilize tandem mass spectrometry within a subsequent vacuum facility to acquire comprehensive molecular structure information. However, recent developments in ambient plasma ionization technology have evidenced that ions can dissociate under atmospheric pressure conditions. As early as 2007, Cooks et al. demonstrated the dissociation of proteins and peptides using thermal APCI[23]. In 2010, Zhang et al. introduced an LTP ion source with two in-source cleavage modes to facilitate structure identification[24]. Furthermore, subsequent studies have reported the capability of liquid-sampling atmospheric-pressure glow discharge (LS-APGD) and arc plasma-based dissociation (APD) to dissociate ions at atmospheric pressure[25,26].

As an atmospheric-pressure plasma technique, microwave plasma is commonly used to perform surface analysis and elemental analysis. Duan et al. developed microwave-induced plasma desorption/ionization (MIPDI), which is a hard-soft ion source that provides a considerable population of fragment ions as well as relatively stable molecular ions[27]. Speer et al. determined the atomic and organic composition of samples applicable to nuclear and conventional forensic screening, including explosive/radionuclide mixtures and inorganic/organic gunshot residue-component mixtures, using high-temperature microwave plasma at 200 W[28]. Microwave plasma below 100 W was reported to detect sterol molecules and attain complete protonation peaks in urban water[29]. In addition, a microwave plasma torch ionization technique, which was performed at 1500 W through a central tube feed, exhibited a hard ionization capability comparable to inductively coupled plasma (ICP), allowing the direct detection of metallic elements in liquid samples[30]. Overall, in microwave plasma-based ionization technology, different power and injection procedures might result in variable ionization effects.

Due to the limitations posed by existing ion sources utilized in ambient mass spectrometry, we propose a wide-energy programmable microwave plasma-ionization mass spectrometry (WPMPI-MS) system. The system can digitally input different function waveforms to rapidly screen plasma ionization energy, thus enabling high-coverage MS analysis of compounds with different properties. This ionization system can achieve similar ionization modes as various mainstream ion sources, including ESI, APCI, EI, and ICP. By accurately manipulating the plasma power through digital means and utilizing pneumatic nebulizer, this method can flexibly achieve soft/hard ionization, in-source cleavage, and element desorption. On this basis, WPMPI-MS can ionize a broad range of molecules, encompassing elements, polar/nonpolar small molecules, and biological macromolecules, with semiquantitative capability. Thus, we anticipate that WPMPI-MS will be an advance ambient-ionization mass spectrometry and an alternative to current commercial ion sources.

## Results

### WPMPI-MS instrumentation and performance

In the WPMPI-MS system, the power of the microwave plasma (0–200 W) can be readily regulated by a ref voltage ($V_{ref}$). A user-defined function waveform with modifiable frequency and amplitude (frequency ≤10 Hz, amplitude within 0–2.5 V) is delivered to $V_{ref}$ as the input signal from a programmable waveform generator to control the plasma power stepwise or without steps (Fig. 1a). The plasma output, for instance, jumps between 80 and 160 W when a step wave is used with particular parameters, such as its input parameter (amplitude of 1.0 V and bias of 2.0 V, Fig. 1b). The frequency of the input waveform is approximately 10 Hz at the fastest, with a maximum energy change rate of 500 W s$^{-1}$, which does not significantly affect the stability of plasma operating state. After the unit stabilizes, microwave plasma torch is characterized by a cylindrical torch 3–8 mm thick and 15–35 mm long. The shape of the torch is influenced by the power and the working gas flow rate (Supplementary Fig. 1). The central material of the argon microwave plasma mainly consists of sub-stable argon and high-energy electrons, while the plasma boundary is filled with various reagent gas ions, including $(H_2O)_nH^+$, $NO^{+}$, and $O_2^{+}$, which are formed by background air ionization[18].

Based on the structure, the plasma ionization energy and gas-phase temperature can be quickly adjusted over a wide range, and we initially evaluated the gas temperature at different plasma power supplies and injection positions (Fig. 1c). The measurement sites are divided into plasma root, core, tip, and 5 mm from the tip (see Supplementary Fig. 1b for specific locations), with the power covering from 60 to 200 W. The spatial distribution of temperature exhibits its peak value at the tip of the plasma, whereas temperatures at the root and core regions are comparatively closer. The overall range in temperature variation covers 350–1300 K.

When the microwave power gradually increased from 60 W to high energy at 200 W, the "ionization softness and hardness" and ionization products changed significantly. Taking methyl salicylate as a model, the signal intensity of the molecular ion ($m/z$ 152.05) gradually increased with increasing plasma power (60–160 W), while the intensity of the protonated ion ($m/z$ 153.05) gradually decreased (Fig. 1d), higher energies (>160 W) result in more cleavage, leading to an overall decrease in signal intensity. The percentage of molecular ion signal intensity versus protonated ion intensity also gradually increases (Supplementary Fig. 2) as the ionization mechanism partially changes from PA (proton affinity) to IP (ionization potential). To some extent the phenomenon may be related to the reagent gas ion species, as we have noted that the populations of reagent ions shift in relative abundance with shifting power (Supplementary Fig. 3). It is evident that the hydronium ions ($(H_2O)_nH^+$), which cause proton transfer, decrease significantly with increasing power; while reagent ion species more likely to cause penning ionization that can bring molecular ions, such as oxygen radicals and nitrogen radical ions, increase in abundance. Notably, whether the ions are protonated is influenced by PA and the functional groups of the compound, with discrepancies between specific compound molecules.

To investigate the capability of WPMPI to perform rapid scanning ionization, we employed various functional waveforms as inputs to ionize reserpine and divalent lead, which have significantly different physicochemical properties (Fig. 2a). Based on the extracted-ion chromatography (EIC) diagrams, the reserpine peaks alternated with divalent lead, and the EIC of lead exhibited better compatibility with the waveform characteristics when rapid scanning ionization of

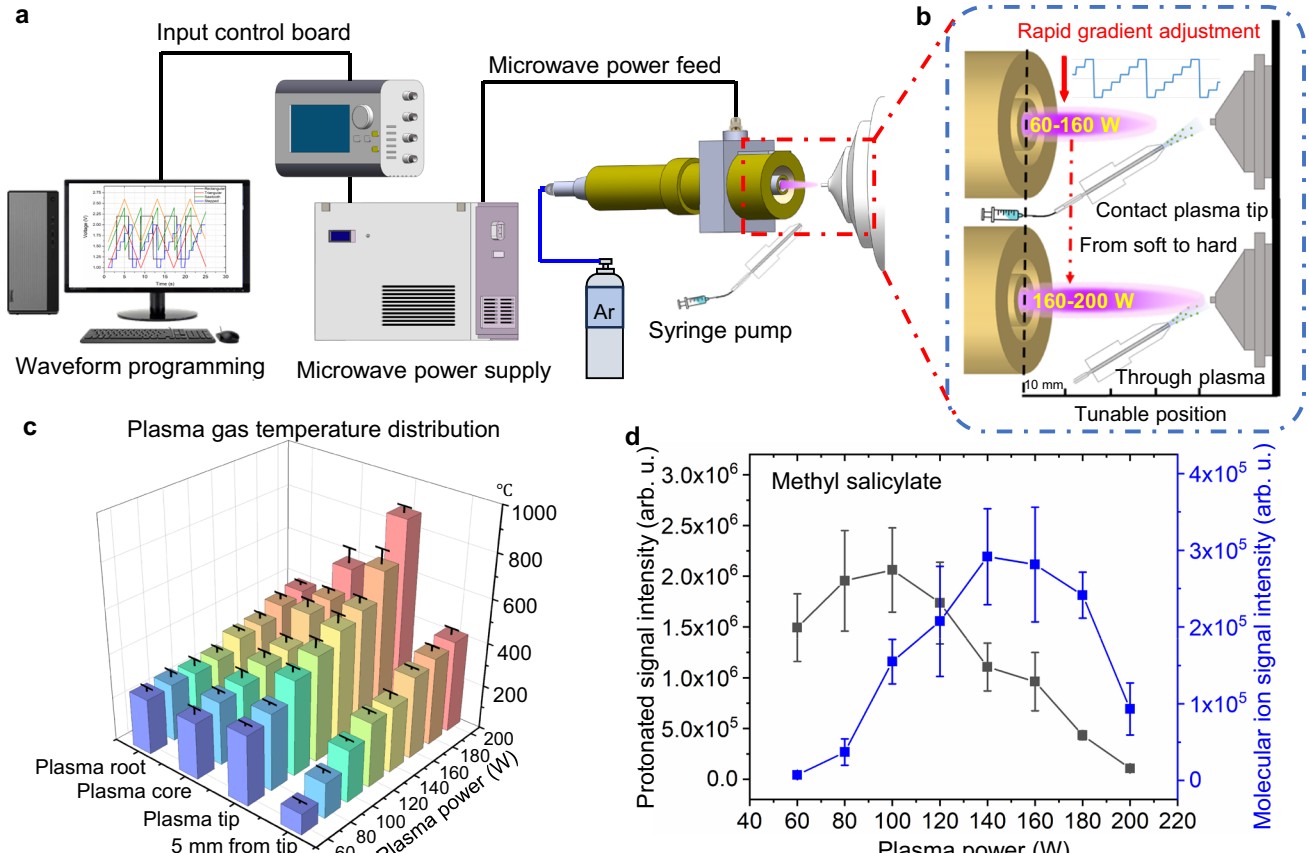

**Fig. 1 | Instrumental setup, principal, performance characterization of temperature and characteristic ions. a** Diagram showing the full setup from left to right: waveform compilation software, control board, microwave power supply, WPMPI source section, mass spectrometer. **b** Schema: the programmable waveform input corresponds to changes in microwave power, causing rapid jumps in plasma morphology and energy, and plumes of spray thus collide with different plasma regions. **c** Histogram of gas-phase temperature at different powers (purple, blue, cyan, green, lime, yellow, orange, and light red stand for 60, 80, 100, 120, 140, 160, 180 and 200 W, respectively) and positions of the plasma. Each value represents the mean ± s.d. ($N = 3$). **d** Ionization products of methyl salicylate: comparison of molecular ion peak intensity (blue) and protonation peak intensity (black) variation with microwave power. Each value represents the mean±s.d. ($N = 5$).

100–200 W was conducted using a sawtooth wave (Supplementary Fig. 4a) and a triangular wave (Supplementary Fig. 4b) at 0.2 Hz. The phenomenon is more pronounced when digitized rectangular-wave scanning ionization is used (Fig. 2b–d) because the microwave power jumps from 100 to 200 W without going through a transition interval. The ionization performance of WPMPI was not affected by the increase in scanning rate within a certain range (<1 Hz).

## Molecular structure and elemental analysis capabilities

To demonstrate the analytical advantages of WPMPI in energy-digitized fast scanning ionization mode, we ionized various types of samples using different energy ranges. We first tested a series of biomolecules in the low-energy region (60–100 W), including peptides, glycans, and oligonucleotides. The ionization products of the Gly-His-Lys (Supplementary Fig. 5a) and bradykinin (Supplementary Fig. 5b) were protonated singly charged ions at $m/z$ 341.20 and $m/z$ 1060.60, respectively, and no significant fragmentation ions were observed. Interestingly, dermaseptin could obtain multicharged, protonated ions in WPMPI-MS (Supplementary Fig. 5c, consistent with the ionization products of commercial ESI (Supplementary Fig. 5f)), which may be due to the action of thermal spray (plasma heat the spray plume) and hydronium ions. The product ions of maltose in WPMPI-MS included the common dehydrogenation negative ion at $m/z$ 505.18, as well as the dehydration negative ion at $m/z$ 487.17 (Supplementary Fig. 5d). Notably, the lower signal response of the glycans in WPMPI-MS may results from dehydration or oxidation of the hydroxyl groups in

the plasma atmosphere. Adenylcytosine cytosine dinucleotide (ApC) was similarly measured in negative mode, showing the dehydrated negative ion at $m/z$ 571.13 (Supplementary Fig. 5e). In addition, maintaining the WPMPI at a low power (60 W) enabled the analysis of some thermally unstable compounds and easily oxidizes compounds. Artemisinin is an important drug for the treatment of malaria, which is thermally unstable because of its dioxygen structure. We employed WPMPI-MS with 60 W energy to analyze the artemisinin solution, abundant protonated molecular ions were observed (40%), indicating that the structure of artemisinin was preserved to a certain extent (Supplementary Fig. 6a). The easily oxidizable compound $N$-isopropyl-$N'$-phenyl-1,4-phenylenediamine (IPPD) was ionized using WPMIP at 100 W, resulting in an abundance of the oxidized products of only 9% and 4%, and the main ionization product was the protonated molecular ion (Supplementary Fig. 6b). Ascorbic acid (vitamin C) was also analyzed using WPMPI at 60 W, and surprisingly no significant oxidation products were observed (Supplementary Fig. 6c), with the highest abundance of ionization products being deprotonated ions.

Setting WPMPI-MS with an energy scanning interval of 140–180 W, the sample plume output from the nebulizer passes completely through the high-energy microwave plasma torch, enabling hard ionization akin to electron impact. Polycyclic aromatic hydrocarbons (PAHs) are common persistent organic pollutants (POPs) that cannot be ionized by ESI. When using EI sources, the ionization products of PAHs are radical ion species formation, [M]·+, while using commercial APCI sources exhibit a high abundance of protonation[31], [M + H]+.

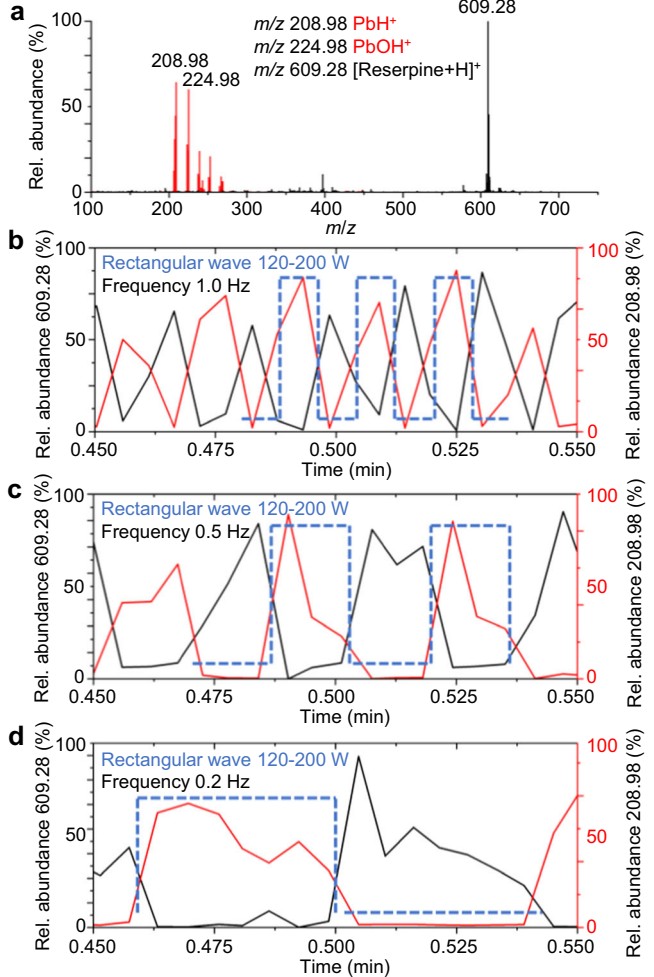

**Fig. 2 | Mass spectra and extracted-ion chromatography (EIC) plots of reserpine and divalent lead when ionized with different scanning waveforms (120–200 W, blue line). a** Mass spectra of divalent lead ions (red) and reserpine ions (black) detected simultaneously under rectangular wave scanning. EIC plots for both ions when rectangular wave at (**b**) 1.0, (**c**) 0.5, and (**d**) 0.2 Hz was applied, the red line represents the EIC plot of divalent lead ions, and the black line is the EIC plot of reserpine, the blue dashed line represents the applied rectangular wave timing.

Four representative PAHs (anthracene, pyrene, chrysene, and benzo(a) pyrene) were analyzed by WPMPI-MS, and the product ions were high abundance of radical molecular ions [M]·⁺ (Supplementary Fig. 7). The response spectrogram in WPMPI-MS is consistent with the EI mass spectra of the compounds in the NIST (National Institute of Standards and Technology) research library. Besides PAHs, WPMPI-MS above 140 W can readily ionize low/non-polar compounds that are undetectable by ESI, such as methyl phenyl sulfide and ferrocene (Supplementary Fig. 8). In addition, we observed cleavages and rearrangements in WPMPI-MS that commonly occur in EI sources. For example, anthraquinone lost small molecules (one or two CO, $m/z$ 181.06 and $m/z$ 153.07) under WPMPI, which is also a common elimination reaction in EI mass spectra (Supplementary Fig. 9a). McLafferty rearrangement was also observed in WPMPI at atmospheric pressure. During this process, ethyl acetate underwent McLafferty rearrangement to eliminate ethylene, resulting in a high abundance of the product ion $m/z$ 61.03 (Supplementary Fig. 9b). 2-pentanone and 4-phenylbutylamine were used to conduct rearrangement reactions in WPMPI to eliminate the effect of direct cleavage (Supplementary Figs. 10, 11). The rearrangement product of 4-phenylbutylamine (an aromatic compound) was the benzyl cation with radicals at $m/z$ 91.05,

while pentanone showed a rearrangement product ($m/z$ 59.05) similar to ethyl acetate (Supplementary Fig. 12). The results may have potential implications for studies on gas-phase ion reactions and plasma organic synthesis at atmospheric pressure.

Several studies confirmed that microwave plasma-based ionization technology has the same elemental analysis capability as ICP[30]. While in WPMPI-MS, the advantage of simultaneous detection of organometallic compounds and metal elements in real time can be achieved by scanning the plasma power. Tributyltin hydride was ionized using a gradient power scan of WPMPI-MS, and its ionization products at 140 W included the molecular ions of reduced hydrogen at $m/z$ 291.11, the desorbed metal elements at $m/z$ 120.91, and the ions of debutylated at $m/z$ 178.99 and $m/z$ 235.05 (Fig. 3a). The composition of ionic compounds can be clearly characterized by high-resolution mass spectrometry and isotope peaks (Supplementary Fig. 13). Isotope mass spectra with matched abundances were also observed using a 200 W WPMPI to directly analyze a solution of 10 ppm lead chloride (Supplementary Fig. 14). The modulation of microwave plasma power can effectively regulate the ionization pattern, as the real-time variation of the product ions was observed during the scanning of triphenyltin chloride (Fig. 3b). Highly abundant chlorotriphenyltin dehydrogenated molecular ions $m/z$ 351.02 were observed at 80 W power, without the relevant peaks of desorbed tin. With increasing scanning power, the elemental and molecular peaks exhibited close relative abundance at 140 W ([SnNOH]⁺ $m/z$ 150.92, [SnOH]⁺ $m/z$ 136.91), and as the power reached 200 W, the molecular ions of compound dehydrogenation completely disappeared. Additionally, the ionized product of desorbed tin elements ($m/z$ 150.92, $m/z$ 136.91, $m/z$ 120.91) exhibited higher relative abundance. In addition, WPMPI demonstrated reliable sensitivity for the detection of inorganic heavy metals (Supplementary Fig. 15, in water: methanol (1:1)), but is slightly inferior to ICP-MS. The result may provide further opportunities for the speciation of organometallic and inorganic metal element in environmental pollution. Furthermore, WPMPI-MS can desorb individual halogen elements in negative ion mode, and a clear isotopic peak of bromine monomers (from dissociated hexabromobenzene) was observed (Supplementary Fig. 16).

Based on this fast and wide energy-range scanning ionization mode, WPMPI-MS enables controlled in-source dissociation to help reveal compound structures, which is demonstrated using methyl salicylate as a model compound. The results showed that the fragment ions of methyl salicylate changed in species and abundance with increasing scanning power from 80 to 200 W (Supplementary Fig. 17). Among the fragment ions, the even-electron ions ($m/z$ 121.03, $m/z$ 153.05) are identical to their ESI-CID spectrums, and the three odd-electron ions $m/z$ 92.03, $m/z$ 120.03, and $m/z$ 152.05 are consistent with the EI spectrum (Supplementary Fig. 18), which is consistent with the fragmentation pattern of arc plasma-based dissociation (APD)[26]. We speculate that the ionization mechanism gradually shifts from "ESI-like" to "EI-like" with increasing power, leading to the transfer of dissociated fragments. Scanning microwave power has a significant effect on the abundance of fragments in WPMPI-MS, peak at $m/z$ 313.16 is the protonated ion of the dipeptide (Phe-Phe), and the percentage of fragment ion at $m/z$ 120.08 rose with increasing microwave power, while the abundance of two fragment ions, $m/z$ 166.09 and $m/z$ 203.12, first increased and then decreased (Supplementary Fig. 19, the speculation of fragment ions in Supplementary Fig. 20).

Since ribonucleic acids and peptide chains are essential life-supporting components, mass spectrometry fragmentation is a key technique for elucidating their structure and sequencing. We demonstrate the utility of WPMPI-MS for biomolecule structure inference using examples of Gly-His-Lys and ApC. ApC was observed to cleave at multiple sites in WPMPI-MS, yielding more information on the fragment ions obtained (Fig. 3c). In addition to the dissociation of some conventional sites ($m/z$ 328.05), the fragmentation pattern of

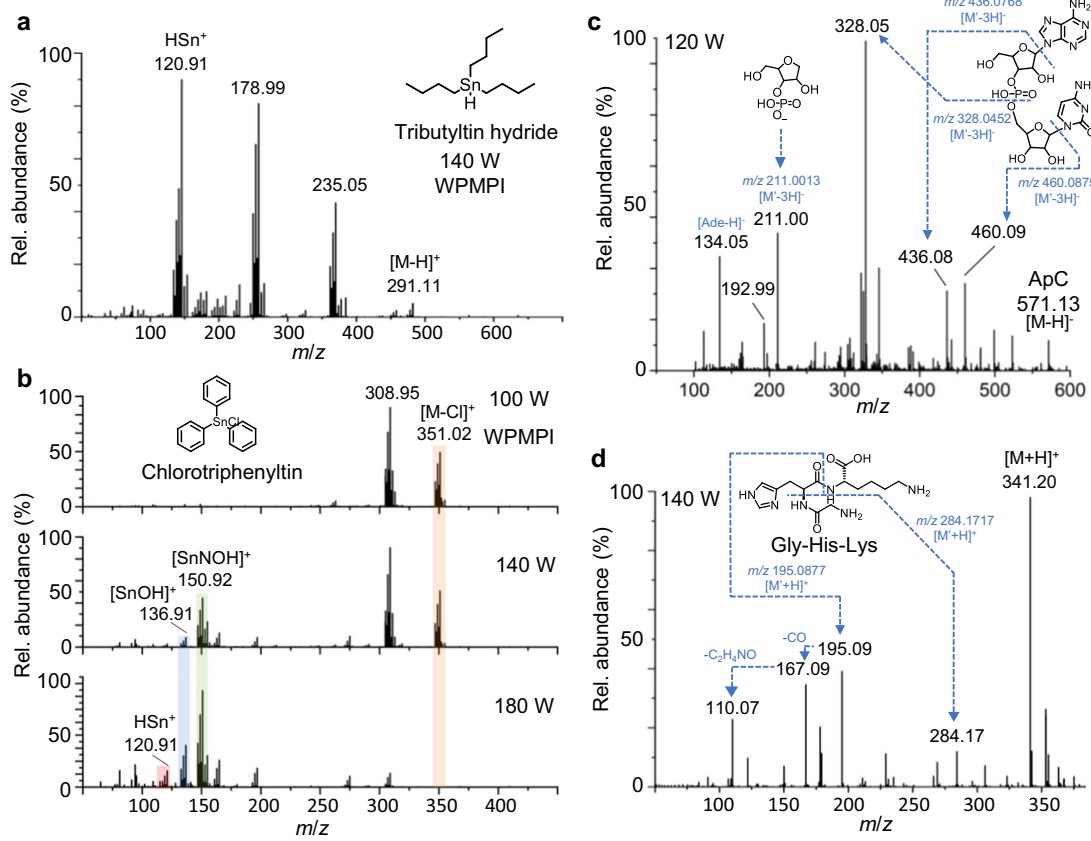

**Fig. 3 | WPMPI-MS provides metal-element analysis capability and biomole-cular in-source cleavage capability. a** Mass spectrum of tributyltin hydride in WPMPI-MS at 140 W. **b** Mass spectra of chlorotriphenyltin at different powers during fast energy scanning in WPMPI-MS (orange rod: mother ion; green, blue, red rods: daughter ions). **c** In-source dissociation spectrogram of ApC. **d** In-source dissociation spectrogram of Gly-His-Lys.

WPMPI-MS generated a certain abundance of ions stripped of bases ($m/z$ 460.09, $m/z$ 436.08, $m/z$ 134.05), which may facilitate sequence identification. The dissociation pattern of Gly-His-Lys under this method differs from that of CID, but fragmentation is abundant, and the ionic structure can be simply deduced (Fig. 3d, $m/z$ 284.17, $m/z$ 195.09, and $m/z$ 167.09 for the dissociation of amide bond CO-NH). The results illustrate that the method can be used for controlled in-source cleavage of ambient fingerprint mass spectra.

In addition, the reproducibility of the signal or fragmentation patterns of WPMPI-MS was investigated by selecting two compounds and testing their fragmentation ratios at four different power steps. The recordings were taken weekly for a cumulative total of 28 days, and the day-to-day relative standard deviation (RSD) was calculated. WPMPI demonstrated good reproducibility for different power steps and analytes (Supplementary Fig. 21). The vertical coordinate represents the fragment ion to parent ion intensity ratio, and the day-to-day RSD of the four fragment ion ratios ranged from 5.2% to 13.5%.

### Application scenarios

To demonstrate the ability of the technique to rapidly analyze trace samples, we tested nine toxicological standards added to a drop of serum. The samples included pesticides (anilofos), PAHs (anthracene, benzo[a]pyrene), inorganic heavy metals (divalent lead), metal organics (chlorotriphenyltin), and psychotropic and conventional drugs (sulpiride, midecamycin, prochlorperazine, reserpine). After the samples underwent simple deproteinization, the nine compounds with widely differing physicochemical properties could be ionized directly using a 100–200 W stepped-wave scanning mode (Fig. 4a). During this scanning process, the analytes could respectively meet their optimal energy position; therefore, each compound has the highest signal

response at one energy step during scanning (Fig. 4b). Utilizing digi-tized step-waves scanning of the plasma power ensures that the signal responses of different compounds can reach the highest sensitivity under the WPMPI-MS system. In practice, the energy-scanning range can also be adjusted according to sample characteristics. To demon-strate the analytical power of WPMPI-MS, we used digital scanning ionization to directly quantify these nine analytes in serum, which showed a good $R^2$ (0.9355–0.9996) over a linear range of 10–5000 μg L$^{-1}$ (Supplementary Fig. 22). The limit of detection (LOD) and limit of quantification (LOQ) ranges were 0.88–14.49 μg L$^{-1}$ and 2.94–48.31 μg L$^{-1}$ (Supplementary Table 1), respectively. In addition, since the scanning time of WPMPI-MS is in the range of 0.5–5 s, the device can be quickly coupled with liquid chromatography to ensure that the compounds within a liquid-phase peak are highly ionized. Within a substance peak retention time, the method can complete multiple energy scanning processes, generating dense small peaks in the EIC diagram (Fig. 4c). By fitting the highest point of each peak and calculating the area, quantitative analysis is possible.

To highlight the concept of digitization and to validate the ioni-zation benefits of WPMPI fast scanning waveforms, we analyzed spiked soil samples using WPMPI stepped waves. Three perfluoroalkyl com-pounds (perfluorobutane sulfonic acid (PFBS), potassium perfluoro-1-octanesulfonate (PFOS-K), pentadecafluorooctanoic acid (PFOA)) and cadmium stearate were added to the soil extraction solution to for-mulate a mixture of 1 mg L$^{-1}$. The samples were analyzed by WPMPI using a stepped wave of 80–200 W (<30 s). Four samples were mea-sured at different power steps: PFOA, which is prone to cleavage, showed the highest signal intensity at 80 W and could not be seen above 120 W; PFOS-K and PFBS are similar in chemical properties and more stable, had an optimal ionization power of 160 W; and cadmium

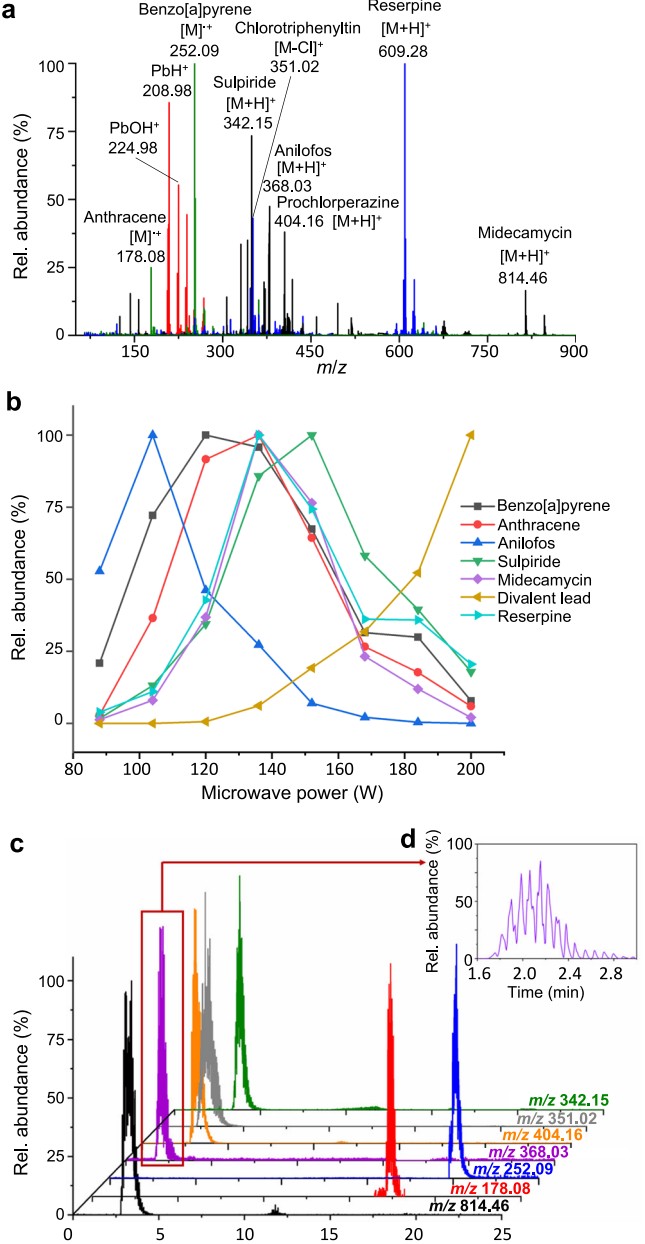

quartz capillary and placed directly on the plasma tip, resulting in stable and persistent mass spectra signals within the mass range of $m/z$ 300–700 (Supplementary Fig. 23). Fourteen highly abundant ion signals were screened out and preliminary identified as representative diterpenoids in aconitum by high-resolution mass spectrometry (Supplementary Table 2). In addition, we performed non-targeted analysis of the landfill leachate by LC-WPMPI-MS in stepped-wave energy scanning mode (100–200 W), and MS data acquisition in full scan with data-dependent acquisition (FS-DDA) mode. More than hundreds of ions were detected, and 56 ions were initially categorized (Supplementary Table 3), including metal elements, low/non-polar VOCs, metabolites, food additives, natural products, pharmaceuticals, and industrial chemicals, demonstrating the ability of the WPMPI-MS in conducting non-targeted analysis of actual samples.

## Discussion

WPMPI-MS offers advance coverage of analyte species compared to conventional ion sources (Supplementary Table 4), and is also electrically simpler than the complex combined AMS technique. Digital, fast-switching, high-coverage mass spectrometry allows for minimal sample consumption in the analysis of complex samples. In a follow-up study, a microwave power supply with higher power could enhance the elemental analysis ability of this method. On the other hand, the method commonly uses a nebulizer to feed the sample, which can be integrated as an electrospray ion source by adding high voltage to the nebulizer selectively, thus introducing analytical capabilities for native biomolecules. In addition, we discussed the safety of using microwaves in ambient ionization sources (Supplementary Fig. 24). We applied perfect shielding to both the power module and the transmission line, and used professional RF connecting lines to place microwave leakage. Microwave leakage is as low as 0.78 mW cm$^{-2}$ at 10 cm of the plasma tube, which is undetectable from 20 cm away, allowing the safety of the operator to be guaranteed.

In summary, we developed a wide-energy programmable microwave plasma-ionization mass spectrometry system with multiple energy scanning modes. A variety of samples with significant differences in physicochemical properties were used to demonstrate the good ionization capability of the method. The different digitized energy scans of this technology allow for high-coverage ionization of biomolecules, drug molecules, nonpolar compounds, metallic elements and metallo-organics, along with controlled cleavage capability. The WPMPI-MS system performed well in the analysis of real samples, rapidly analyzed nine toxicological standards in one drop of serum and demonstrated good quantification and LC coupling capability. Furthermore, the capability of WPMPI to concurrently detect PFAS and heavy metal contamination was validated through spiked soil samples, while the direct analysis capacity for solid samples with complex matrices was tested using Yunnan Baiyao powder. The ability of WPMPI-MS to cope with non-targeted analysis of actual samples was also verified using landfill leachate, demonstrating the wide coverage capability of the method. The technology analyzes a much wider range of compounds than existing ion sources, and the programmable energy scanning mode leads to more flexibility and functionality during analysis. Based on these advantages, we anticipate that the technology will be widely utilized in analytical chemistry and provide an alternative way to address the scalability problem in commercial mass spectrometry.

## Methods

### Materials

Methyl salicylate (99%) and ethyl acetate (99.9%) was purchased from J&K (Shanghai, China). Anthracene (98%), pyrene (98%), chrysene (98%), benzo[a]pyrene (98%), anthraquinone (98%), reserpine (99%), methyl phenyl sulfide (99%), ferrocene (98%), prochlorperazine (95%), and

**Fig. 4 | Analysis of the toxicological samples in serum. a** Superimposed mass spectra after energy scanning ionization (red spectrum: 200 W, green spectrum: 150 W, blue spectrum: 130 W, black spectrum: 120 W,) of spiked serum (including benzo[a]pyrene, anthracene, anilofos, sulpiride, midecamycin, prochlorperazine, chlorotriphenyltin, reserpine and divalent lead). **b** Different analytes (benzo[a]pyrene (black), anthracene (red), anilofos (blue), sulpiride (green), midecamycin (purple), reserpine (cyan) and divalent lead (yellow)) reach the maximum signal response at different energy step positions. **c** EIC diagram (green: $m/z$ 342.15, gray: $m/z$ 351.02, orange: $m/z$ 404.16, purple: $m/z$ 368.03, blue: $m/z$ 252.09, red: $m/z$ 178.08, black: $m/z$ 814.46) of serum samples analyzed by LC-WPMPI-MS. The method can complete multiple energy scans in a single liquid chromatography peak, and the inset picture (**d**) is a magnified anilofos EIC plot near 1.6–3.0 min.

metal, which needed to raise the power to 200 W before a clear isotopic signal could be observed (Fig. 5).

To enhance the evidence regarding the certain anti-matrix interference capability of WPMPI, we conducted WPMPI-MS analysis of principal components in Yunnan Baiyao powder under 100–140 W conditions. A small amount of powder samples was immersed into a

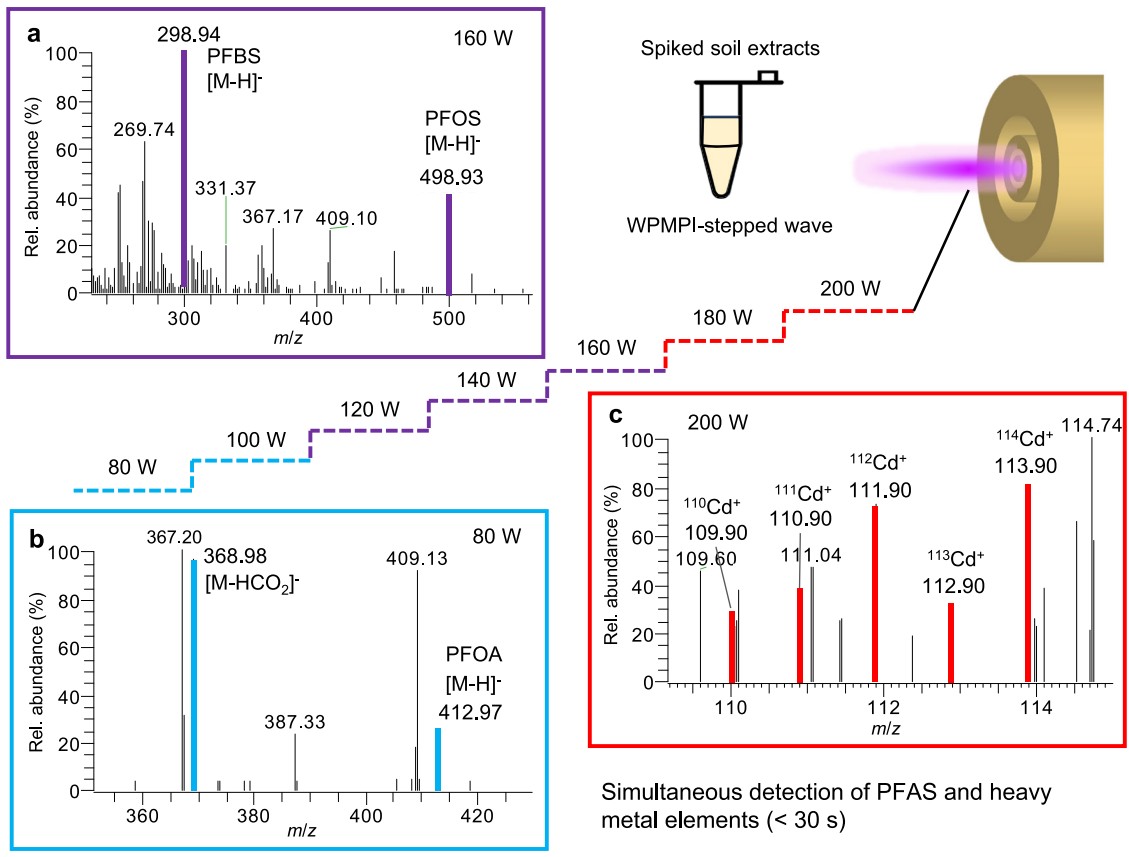

**Fig. 5 | Detection of spiked soil extracts using WPMPI stepped wave power scan mode. a** WPMPI mass spectra of PFBS and PFOS-K at the step of 160 W (purple). **b** WPMPI mass spectra of PFOA at the step of 80 W (blue). **c** WPMPI mass spectra of cadmium at the step of 200 W (red).

midecamycin (98%) were purchased from Aladdin (Shanghai, China). Adenylyl cytosine dinucleotide (ApC, 95%) was provided by Takara Bio Inc. (Beijing, China). Divalent lead (99%), chlorotriphenyltin (96%), tributyltin hydride (97%), and hexabromobenzene (99%) was purchased from Meryer (Shanghai, China). Artemisinin (97%) and *N*-isopropyl-*N*′-phenyl-1,4-phenylenediamine (IPPD, 95%) were purchased from Bidepharm (Shanghai, China). L-Phe-L-Phe (98%), 4-phenylbutylamine (98%), maltose (98%), pentanone (99%), glycyl-L-histidyl-L-lysine (Gly-His-Lys, 98%), anilofos (98.5%), cadmium stearate (98%), bradykinin (97%) and dermaseptin (97%) were purchased from Chentong (Hangzhou, China). Perfluorobutane sulfonic acid (PFBS, 99%), perfluorooctanoic acid (PFOA, 99%), potassium perfluoro-1-octanesulfonate (PFOS-K, 99%), and ascorbic acid (99%) were purchased from Macklin Inc. (Shanghai, China). Yunnan Baiyao powder was purchased from Yunnan Baiyao Group Co., Ltd. Topsoil collected from Sandun, Hangzhou, China. Methanol (HPLC), acetonitrile (HPLC) and water purchased from Merck (Darmstadt, Germany) were used for preparing the sample solvents.

## Experimental

For the preparation of standards, PAHs were dissolved in acetonitrile and the rest were dissolved in water: methanol (1:1). In the standard test experiments, as well as in-source cleavage experiments, a concentration of 10 mg L$^{-1}$ was used for all samples. Injecting the sample through the nebulizer, one analysis time period is less than 3 min; when coupled with LC, it depends on the chromatography process. When analyzing solid samples (pharmaceutical powder), use a quartz capillary tube to dip a few minigrams and place it directly in the plasma area, the analysis time period is less than 30 s.

Chicken serum was purchased from Genom Bio (Hangzhou, China) and prepared as follows. Briefly, 200 μL of chicken serum and 10 μL of a mixture of 9 standards (1 mg mL$^{-1}$) were mixed, to which 800 μL of acetonitrile was added to precipitate proteins. Then, the solution was sonicated for 5 min then centrifuged at 4193 × *g* for 3 min, and allowed to settle at 4 °C for 10 min, and the supernatant was filtered and finally stored at −20 °C before use. The concentration of the spiked serum extract to be measured was 10 mg L$^{-1}$. Blank serum was processed in the same way. A concentration gradient of 0.01–5 mg L$^{-1}$ was configured for quantification. The LOD and LOQ of this method was calculated using the following Eq. (1) (signal to noise =3, 10) based on measurements of spiked samples:

$$\text{LOD} = \frac{3c\sigma}{S}, \text{LOQ} = \frac{10c\sigma}{S} \quad (1)$$

Where *c* is the analytes concentration in the spiked sample, $\sigma$ is the standard deviation of all the measurements (*n* = 7), *S* is the mean value of the signal intensity.

Soil extract preparation process: weigh 4.0 g of soil sample into a 50 mL centrifuge tube, add 10 mL of extraction solution (water: methanol (1:1) solution, containing 1 M HCl), vortex for 1 min and ultrasonic at 40 °C for 1 h. Centrifuge at 1509 × *g* for 10 min, then collect the supernatant. Repeat the above steps, combine the two supernatants and adjust the pH with ammonia to neutral, then add the standards after passing through a 0.45 μm filter membrane to obtain a 1 mg L$^{-1}$ spiked soil extract for measurement.

A raw landfill leachate sample was collected in May 2024 from a domestic landfill in northern Zhejiang, China, using polypropylene vials to collect leachate from the spigots. The samples were transported to our laboratory and stored at −20 °C before preparation. Liquid samples were diluted with an equal volume of acetonitrile, sonicated for 2 min, and then filtered through a 0.22 μm nylon membrane filter, resulting in 4 mL of clarified stock solution before

instrumental analysis. An aqueous acetonitrile solution of the same proportion was prepared as a blank.

The temperature is measured by a thermocouple pyrometer (Mini Type K/J Input Thermometer, UNI-T Co. China) with a range of −50 to 1300 °C. During the test, the temperature probe was placed at the target plasma position for 3 s, the gas temperature value was recorded and averaged over three repetitions. Microwave radiation was detected by a handheld microwave leakage detector (HT-M2 Microwave Leakage Detector, Dongguan Xintai Instrument Co., Ltd.).

## WPMPI instrumentation

The device consists of a wide-energy programmable microwave plasma torch generator, a multi-axis moving platform, a quartz nebulizer and a control board. The microwave plasma generator and a 3D printed dual-axis sliding rail are placed on top of the $xyz$-axis fine adjustment stage, and the nebulizer is placed on the dual-axis sliding rail with its center level with the plasma torch (Supplementary Fig. 25). The quartz nebulizer can be adjusted by a dual rail sliding platform (30 mm × 25 mm rectangular) within the microwave plasma torch area. Liquid sampling experiment were carried out using pneumatic nebulization (quartz nebulizer with concentric structure), using $N_2$ as carrier gas (1000–3000 mL min$^{-1}$). The sample solution was injected into the quartz nebulizer by a syringe pump. The microwave plasma generator is a resonant cavity design consists of microwave outer tube, central tube, and intermediate tube, where the microwave energy forms a maximum electric field and excites the plasma at the end plane of the cavity. To enable fast and stable switching of microwave energy, modifications were made to the resonant cavity and microwave plasma tube. The increased size improves heat dissipation, supports a wider power range, and allows for better impedance matching adjustment. Argon was chosen as the working gas due to its lower discharge power, enabling activation of a stable plasma torch with around 40 W of microwave power. Real-time monitoring of reflected power helps minimize impedance matching issues. A circulator was added to enhance the stability of the power supply's fast switching operation by directing reflected power away from the power supply, and the details are shown in Supplementary Fig. 26, and the microwave-energy utilization reaches over 95%. Argon as the working gas was introduced from the intermediate and central tubes to generate plasma and modify the plasma jet shape. The working gas is controlled by rotameters, adjustable in the range of 500–2000 mL min$^{-1}$ (commonly in 1000 mL min$^{-1}$). The control board is a common microcontroller that can be programmed to control specific-function waveform outputs. The output of the control board is connected to the power supply of the microwave generator, which is used to manage the change of power and timing of the microwave output.

Based on the design, 0 to 200 W adjustable microwave (operating frequency of 2.45 GHz) through a coaxial cable into the intermediate tube, and in the cavity between the intermediate tube and the input tube propagation, to form a stable microwave electromagnetic field. During operation, the plasma needs to be excited by touching the intermediate tube with a metal tip (this process is only applied in the beginning and is removed immediately after plasma activation, the tip does not need to be grounded, and no specific material requirements). The angle between the tip of the nebulizer and the plasma torch is 60°, and the focal-point position is directly opposite and adjustable from 0–25 mm to the mass spectrometer inlet. Thus, the flexible multi-axis platform allows the sample to be fed into different regions of the plasma, and a programmable plasma-power energy range permits differences in ionization energy and temperature. It is worth noting that argon is necessary and that the use of helium or nitrogen will result in a high plasma excitation power, which will not allow for a lower plasma temperature interval.

## Liquid chromatography and mass spectrometry methods

The liquid chromatograph coupled WPMPI is a Thermo Fisher UltiMate 3000 UPLC. The chromatographic column was an HYPERSIL GOLD C18 column (2.1 mm × 100 mm, 3.0 μm), and the gradient elution was performed using mobile phase A (1‰ formic acid in water) and mobile phase B (acetonitrile). Flow rate: 0.4 mL min$^{-1}$; injection volume: 3 μL; column temperature: 30 °C; elution gradient program: 0.0 min, 10%B (10% acetonitrile in water); 0.0-20.0 min, 10-100%B; 20.0-30.0 min, 100%B; 30.0-31.0 min, 100-10%B; 31.0-35.0 min, 10%B.

The experiments were carried out using a Thermo LTQ Orbitrap mass spectrometer (Thermo Fisher Scientific, Waltham, MA). Data acquisition was performed using the Xcalibur® software (version 2.2 SP1.48; TFS, SanJose, CA) embedded in the instrument. The instrument settings were as follows: capillary voltage, 30 V; capillary temperature, 300 °C; tube lens voltage, 100 V. microscan: 3. The precise and contrasted mass analysis was calibrated using the following ESI parameters: sheath gas (nitrogen) flow rate, 6 arb. u.; aux gas (nitrogen) flow rate, 0 arb. u.; capillary temperature, 275 °C; spray voltage, 4.5 kV; capillary voltage, 30 V; tube lens voltage, 110 V. High-resolution mass errors for all analytes are shown in Supplementary Table 5. The reagent gas ions detected by a home-made time-of-flight mass spectrometry (assisted development by Suzhou Zhipu Weixing Intelligent Technology Co.).

A Orbitrap Exploris 120 (Thermo Scientific, Bremen, Germany) mass spectrometer was used in conjunction with the WPMPI to guarantee the accuracy of the non-targeted analyses when analyzing the landfill leachate. Data acquisition was operated in the negative or positive ionization mode using full scan with data-dependent acquisition mode (FS-DDA). The instrument settings were as follows: capillary temperature, 320 °C; RF lens, 70%. Each FS-DDA loop contains one MS$^1$ scan plus four MS$^2$ scans. The method parameters were set as follows: MS$^1$ $m/z$ range, 70–1000; fragmentation of parent ions with signal intensity higher than $5 \times 10^5$; mass selection range, 0.5 g mol$^{-1}$; collision energies, HCD 50. During data processing, FreeStyle v1.8 (Thermo Scientific, USA), and the NIST database were used for comparison of molecular formulae. FS-DDA data were analyzed using MSFinder v3.6[32,33], and the ions that meet following conditions in the dataset are considered to be detected: MS$^2$ scores > 5.0, signal intensity >$5 \times 10^5$, mass error < 10 ppm.

## Reporting summary

Further information on research design is available in the Nature Portfolio Reporting Summary linked to this article.

## Data availability

The data that support the findings of this study are available from Figshare[34] and from the corresponding authors upon request.

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

## Acknowledgements

We thank Prof. Cuirong Sun at Zhejiang University, Prof. Ping Cheng at Shanghai University and Dr. Lingfeng Li at Suzhou University for their helpful discussions. This work was supported by National Natural Science Foundation of China (21927810 (Y.P.), 62174147 (X.W.), 22336004 (Y.P.)) and the Key Research and Development Project of Zhejiang (2022C01141 (X.W.)).

## Author contributions

F.C. performed the conceptualization, data curation, experimentation, investigation, methodology, writing the original draft, and editing. G.Z. participated in the methodology and investigation. W.W. did the formal analysis and validation. N.S.S. did the language polishing. X.W., Y.P. and H.F. participated in the supervision, conceptualization, and resources and funding acquisition.

## Competing interests

The authors declare no competing interests.
