## [Peer Review File · Nature Communications]

Wide-energy programmable microwave plasma ionization for high-coverage mass spectrometry analysisEditorial Note: Parts of this Peer Review File have been redacted as indicated to remove third-party material where no permission to publish could be obtained.

REVIEWER COMMENTS

Reviewer #1 (Remarks to the Author):

The manuscript by Wang and co-workers presents a somewhat novel approach to the ionization of diverse chemical species using a single ionization source. Indeed diverse species are ionized to differing extents, but the manuscript as a whole (even as a Communication) lacks sufficient detail and clarity to allow others to duplicate the experiments or to demonstrate that the authors have an understanding about the overall topicality in terms of plasma-based ionization sources. There is much hype, including the title, but also some encouraging results. In total, the quality of the submission does not meet the standards of a Nature Communications. Some of our specific comments follow:

1. The use of the term "digital" in the title is misleading. In most uses of the phrase, digital refers to a 1-or-0 situation; all-or-nothing, all of one situation and then one opposite one. In the context of this work, it would imply plasma operation conditions where only-molecular or only-atomic ions are formed; these situations are not demonstrated.
2. The authors use the term "ambient mass spectrometry" very loosely. One source defines the area as: a form of ionization in which ions are formed in an ion source outside the mass spectrometer without sample preparation or separation. Another defines it as: analyzing samples as they exist in the real world. In most instances, this refers to materials in the solid state wherein some form of desorption is applied. Here, solutions are introduced via a nebulizer (perhaps) and in some examples through a liquid chromatograph. The device described here is simply a plasma operating at atmospheric pressure.
3. Ln 47-49: Here, the authors list the means that ambient MS is implemented to remove species from a sample surface, perhaps suggested (though poorly) in the phrase "to directly detect".
4. Ln 56: To which 3 mass spectrometers are the authors referring?
5. Ln 81: The authors refer to the liquid sampling atmospheric pressure glow discharge source, ref 25. Indeed, this group has published a multitude of papers illustrating the concept of the combined atomic and molecular (CAM) ionization source. We believe that this is the concept to which the authors seek to address. The authors would do well to become familiar with this literature.
6. Ln 91-94: Neither the previous reports of MIP or ICP sources "allow the dissociation of charged metallic elements". This is phenomenologically incorrect and we again encourage the authors to be more knowledgeable in the literature of plasma source mass spectrometry.
7. Ln 102-103: We do not at all understand the term "atomization injection". No form of nebulization (which is what is being done here) introduces free atoms. Indeed, throughout the manuscript, including the Methods section, the description at times suggests and electrospray source and in other a pneumatic nebulizer. This lack of clarity is confusing and suggests a lack of understanding on the part of the authors.
8. Ln 113: "Any frequency and amplitude"? Surely there are some limitations.
9. Ln 128: The "intersection" of what and what? Here again, the regions of plasma structure and sampling are very well defined in the relevant plasma literature.
10. Ln 129-131: What is meant by "temperature"? Is this the kinetic, ionic, electronic, vibrational, or excitation temperature? They each have a very specific meaning in the realm of plasma chemistry. How was it actually measured? The values presented have absolutely no meaning.
11. Ln 149: We see no "schematic diagram" of the molecular ion peak . . .".
12. All mass spectra of metal containing species: We note that in none of the presented spectra are there actual atomic ions, they appear as the metals with attached H, OH, etc. species. So, the "digital" situation is not realized. More importantly, while not readily apparent by the bare eye, magnification of the spectra reveal isotope abundances that are VERY MUCH in error. Indeed, if one were presenting a new ionization source for atomic species, some reporting of the determined isotopic abundances is required. The works of Marcus et al. in atomic mass spectrometry using Orbitrap analyzers would be very helpful to the authors.
13. Fig. 3 caption and discussion: The provided caption "WPMPPI-MS provides metal analysis with in-source cleavage capability" is a complete misnomer. Only 2 of the compounds contain metals (again

without any atomic ions observed), with the ApC and copper peptide not actually containing metals or indeed being dissociated to that level. Upon our looking on the internet, the term copper peptide is completely misused here. The presented molecule is a copper-binding peptide, but the authors present it as it contains Cu, which clearly is not evidenced in the spectra.

14. Fig. 4a caption and discussion: a) What is/are superimposed mass spectra? If the different species are ionized at different applied powers, how is the data obtained? C) What is an "EIC diagram"? Are we looking at a extracted ion chromatogram (EIC)?

15. Ln 263-267: We are not told the method of computing the LOD or LOQ values for the various analytes. We have problems with response curves performed covering only 2 orders of magnitude in concentration space. If we look at the response curves in Fig. S17, the values for many of the compounds presented in Table S1 are not at all sensible. For example, looking at the curve for benzo[a]pyrene, we see that the curve "bottoms-out" at approximately 150 µg/L, but the table reports an LOD of ~2 µg/L. This cannot be correct by standard means of calculation. Visually, very few of the curves would be expected to yield such low values as the table.

Reviewer #2 (Remarks to the Author):

This manuscript is an exciting report of a new microwave plasma-based ionization source for mass spectrometry that is capable of rapidly modulating the power through application of digital waveforms. This research is an interesting advance in mass spectrometry ionization. This work presents numerous applications of this technology, supporting the claims of the authors. Building on this work could lead to an important new analytical tool in environmental speciation work, forensic science, and materials characterization. An ion source capable of both molecular and elemental analysis is compelling in a wide range of fields. It should be published after revisions. Revisions should primarily focus on providing more experimental details. Specific suggestions are listed below.

I encourage the authors to be more precise in their language surrounding ambient ionization. Infusing samples in solution into an ion source is not generally considered ambient ionization. If the sample is not being introduced from a solid platform between the ion source and the MS inlet, it is not an ambient ionization experiment.

The specific design of the microwave plasma is critical to the data presented and is largely overlooked. It is briefly described in words as some kind of coaxial design, but the description is inadequate. While microwave plasmas are less fickle than inductively coupled plasmas, a specific configuration of a microwave plasma is generally only stable under a very limited range of power and gas conditions. The core of this paper is that the authors have managed to maintain a plasma over a range of energies at a constant gas flow while rapidly fluctuating the power. There must be some innovation in the design of this microwave plasma to accomplish these things. Please provide significant details. Is this a microwave-induced plasma as opposed to a capacitively coupled microwave plasma? Is this a non-resonant design? Did the authors encounter issues with impedance matching and reflected power over this range of plasma conditions? If this is based on a previous microwave plasma design, cite that work and describe the authors' modifications. Provide substantial details and figures/schematics to explain this innovative design at the heart of the exciting data presented in the manuscript.

While the microwave plasma design is the most critical detail, the authors should also elaborate on other experimental details. Specific instances are listed below:

- How was the temperature of the plasma measured? How were the reagent gas ions in the plasma measured? These smaller ions can be difficult to monitor in an LTQ-Orbitrap. Did the authors measure these ions directly with this instrument? Did they use an optical method?
- Did the populations of reagent ions shift in relative abundance with shifting power? This is an important detail, given the observed shifts from protonated molecules to molecular ions that were observed in the spectra with increases in applied power. Presumably, the population of species more

likely to cause penning ionization, like oxygen radicals, would increase with increasing power, while the hydronium ions more likely to promote proton transfer would decrease.

- For the nebulizer used to introduce the sample into the plasma, the authors note that they can introduce 0-5 kV to it. This is a critical detail. If the authors are applying high voltage to the nebulizer, they are doing ESI in addition to any microwave plasma. We would expect many of the spectra presented to appear if only ESI was used (none of the spectra with EI-like fragmentation). The authors should carefully detail when and what voltages were applied to the nebulizer and how that voltage was coupled to it. They should analyze how the voltage applied at the nebulizer influenced the spectra and provide data to support their conclusions.

- Concentrations of samples should be listed for all the data presented. It is provided for some experiments and not for others. Concentrations are explicitly listed for the serum studies but not for the other example analytes listed prior to those experiments.

- Figure S5 shows a mixture of PAHs ionized at moderate powers. There is not significant fragmentation observed in this spectrum. In Figures S6 and S7, there is significant fragmentation. What has changed? Presumably, this is due to an increase in power, but it is quite important that the authors specify this. I would encourage the authors to include the power applied and sample concentration in each of the captions for the spectra in the main body and supplementary information.

- The authors note that during operation of the plasma, a metal tip is applied to the intermediate tube. Is this tip grounded? What is it connected to? Is it applied throughout the experiment, or only in the beginning? What is it composed of, and is any metal contamination from this tip observed in the spectra at high powers?

- I encourage the authors to superimpose the waveforms on the EIC data in Figures 2 and S3. The correlated timing of the applied power and MS signal is an interesting detail that should be commented on.

Reviewer #3 (Remarks to the Author):

In this study, a digitally regulating microwave plasma energy device (named wide-energy programmable microwave plasma ionization, WPMPI) was developed and applied to ionize analytes introduced through nebulizing or electrospraying liquid samples. Different ion types were detected by the WPMPI-MS, including radical ions for PAHs, protonated ions for organic standards and multiply charged ions for large biochemical compounds. Unfortunately, the development of microwave plasma ionization technology (references #27-30 and more) as well as changing the high voltage applied to gas molecules to vary the plasma energy to induce different ionization mechanisms (reference #15 and more) have previously been reported, meaning the novelty of this study is not high. In addition, several substantial inquiries as listed below mean this reviewer is overall unable to support the publication of this manuscript in Nature Communication.

1. This manuscript does not adequately explain how the developed method is novel; specifically, the significance of the formation of multiply charged analyte ions in the WPMPI (P.9, line 173-174 and Fig. S4c) is not explained upon by the authors. During WPMPI-MS analysis, sample solution was introduced into the source through nebulization or ESI mechanisms (p.17, line 324-327). Therefore, WPMPI seems to be a combination of two ionization methods: ESI and plasma APCI. But the concept of dual ionization source has previously been reported in several publications, where plasma or corona discharge APCI was used to ionize low-to-medium polarity compounds (to form radical or protonated analyte ions) and ESI was used to ionize high polarity compounds (to form protonated or multiply charged analyte ions). The plasma energy could be changed simply by varying the voltage applied to gas molecules to induce different ionization mechanisms, with one of the examples being reference #15 in this manuscript.

2. As is generally known, different high energy species are present in plasma including radicals, protons, electrons, and cluster ions. These species can react with the analytes to form different types

of analyte ions. It is then not surprising to see radical and protonated analyte ions on plasma-based WPMPI mass spectra. However, the study would greatly benefit from presenting and comparing mass spectra obtained from EI (in vacuum) and WPMPI (ambient conditions) to better consider the similarities and differences between the two ionization methods.

3. Using microwaves in ambient ionization sources is a major safety concern. Exposure to microwaves can cause mild skin burns or cataracts and a piece of metal can act as an antenna, which can create sparks and start a fire.

4. The concentrations of the standard sample solutions used in this study were not reported at all (see text and Figs. 1-3 and Figs. S3-S15).

5. The authors claim that the ion signals of 9 standards spiked in the serum were detected by WPMPI-MS simply after protein precipitation and with no serious interferences from blood matrices (P. 13, line 255-257 and Fig. 4a). This would not be a logical conclusion unless the concentration of the spiked standards is high. Indeed, my own calculations place the sample concentration to be at a high level - 10 ppm (P.17, line 312-316), thus explaining why the ion signals of spiked standards were detected.

6. The authors claim that the sensitivity for the detection of spiked standards in serum with WPMPI-MS is at sub-ppt level (P. 13, line 264-265). Since the sensitivity of most ambient ionization mass spectrometric technologies is at the ppb level, it would be very difficult to achieve such high detection sensitivities from complex serum samples without additional sample processes like extraction, concentration, and chromatographic separation.

Reviewer #4 (Remarks to the Author):

Authors present an ionization technique that is based on the microwave plasma. The work is based on previously known technology that is applied in a new way. Authors are aiming to digitalize the ionization and also broaden the range of analytes.

In general, the results shown here are promising and could be of great value in the origin of life field.

Major comments:

1. Since the paper is not only a proof of concept anymore and is targeted to show that the technology is applicable for wide range of analytes, the paper is lacking this variability of analytes. Analysing 9 toxicological standards is enough for proof of concept but for this paper application for wider range of analytes should be shown.

2. One of the main drawbacks of the paper is that it is missing characterization of the ionization. For example, it lacks discussion on the reproducibility of the signal or fragmentation patterns.

Investigation of matrix effects would also provide additional information.

3. Since the paper is claiming "unprecedented coverage" compared to conventional ion sources, a better comparison should be presented.

Minor comments:

1. Title refers to "digital ionization". It is unclear what makes this ionization digital on comparison to previous techniques. Reviewer is in the opinion that by using programming for the ionization does not make the ionization digital.

2. Furthermore, it remains unclear what are the "trace biosamples" mentioned in the title of the paper.

3. Work presents LoD and LoQ values but it is not specified how these are calculated.

4. Line 215 - exhibits -> exhibit

5. The overall readability of the paper should be improved, especially sections related to results and discussion.

6. The time period of the analysis has not been specified.

Reviewer #5 (Remarks to the Author):

A wide-energy programmable microwave plasma ionization mass spectrometry (WPMPI-MS) is presented with various characteristics by digitally regulating the microwave energy. This is an interesting modification of the ion source and deserves further investigation. However, I think more work needs to be done before the paper can be published.

1. In the WPMPI-MS system, the power of the microwave plasma (0-200 W) can be regulated by a ref voltage. The overall range in temperature variation covers 350 K-1300 K. It is recommended to increase the investigation of thermally unstable samples. The influence of excess energy produced and active species on the analysis of oxidizes easily compounds should be discussed.

2. A conclusion given by the author is this method can achieve high coverage of substances with various characteristics by digitally regulating the microwave energy. It should be increased whether the coverage of organic compounds is increased, especially whether the response of compounds that are not easily ionized by ESI is increased.

3. How sensitive is this technique for the determination of metal elements compared to ICP-MS, and if the difference is too large, what are the application scenarios for the simultaneous detection of metal elements and organic compounds? This should be illustrated with practical examples. Successful examples of applications are a bit important to highlight this technology.

4. The stability of the analysis results needs to be further improved, which will be the key to the popularization and application of this technology.

Responds to the Reviewers' Comments

Reviewer 1:

The manuscript by Wang and co-workers presents a somewhat novel approach to the ionization of diverse chemical species using a single ionization source. Indeed diverse species are ionized to differing extents, but the manuscript as a whole (even as a Communication) lacks sufficient detail and clarity to allow others to duplicate the experiments or to demonstrate that the authors have an understanding about the overall topicality in terms of plasma-based ionization sources. There is much hype, including the title, but also some encouraging results. In total, the quality of the submission does not meet the standards of a Nature Communications. Some of our specific comments follow:

Comment 1:

The use of the term “digital” in the title is misleading. In most uses of the phrase, digital refers to a 1-or-0 situation; all-or-nothing, all of one situation and then one opposite one. In the context of this work, it would imply plasma operation conditions where only-molecular or only-atomic ions are formed; these situations are not demonstrated.

Responds 1:

Thanks for your advice. We are very sorry for our negligence of the conceptual misleading of the word “digital” in this paper. In fact, what we are expressing with “digital” is not that the plasma operates under conditions that form only molecular or atomic ions. Essentially, the term “digital” as used in the text refers to the changing form of the input power, which we digitally split to form a number (or two) of energy steps, thus enabling a fast ionization process with high coverage of different compounds. In this model, whether or not to ionize a compound can be modulated to some extent, as shown in Fig. 2 and Fig. 5 in the manuscript, this type of ionization that we consider can be called " digital ". Similar to the concept of “digital ion trap”, digital waveform is introduced into the ion trap electrodes, instantaneously switching the frequency and duty cycle of the rectangular waveform to excite the ions instead of converting the amplitude [Int. J. Mass Spectrom. 221 (2002), 117–138. Anal. Chem. 78 (2006), 1995-2000]. Taken together, we plan to change the title to: “Toward

digital programmable ionization: Wide-energy programmable microwave plasma for high coverage mass spectrometry analysis of various compounds”

Comment 2:

The authors use the term “ambient mass spectrometry” very loosely. One source defines the area as: a form of ionization in which ions are formed in an ion source outside the mass spectrometer without sample preparation or separation. Another defines it as: analyzing samples as they exist in the real world. In most instances, this refers to materials in the solid state wherein some form of desorption is applied. Here, solutions are introduced via a nebulizer (perhaps) and in some examples through a liquid chromatograph. The device described here is simply a plasma operating at atmospheric pressure.

Responds 2:

Thanks for your advice. The definition of the term “ambient mass spectrometry” as we understand it is more generally. As defined by Prof. R. Graham Cooks, ambient ionization refers to “the ionization of unprocessed or minimally modified samples in their native environment, and it typically refers to the ionization of condensed phase samples in air.” [*Ambient Mass Spectrometry. Science 2006, 311, 1566-1570.*]. And in Facundo M. Fernandez's review, ambient mass spectrometry is defined as having the following characteristics: (a) ionization in the absence of enclosures such as those typically found in ESI. In other words, the technique should operate in the open air or ambient environment. (b) Ambient MS techniques allow direct ionization with minimum sample pretreatment such as preconcentration, extraction, derivatization, dissolution, or chromatographic or electrophoretic separation. Although this requirement can be relaxed to some extent in challenging applications of ambient MS. (c) Should be interfaceable to most types of mass spectrometers fit with differentially pumped atmospheric pressure interfaces, without substantial modification to the ion transfer optics or vacuum interface. (d) Should generate ions softly. [*Ambient Sampling/Ionization Mass Spectrometry: Applications and Current Trends. Anal. Chem. 2011, 83, 4508–4538.*] Over the past 15 years, innovations in the field of environmental ionization mass spectrometry have evolved rapidly, allowing these techniques to progressively outgrow their concepts. [*Ambient Ionization Mass Spectrometry: Recent Developments and Applications. Anal. Chem. 2019, 91, 4266–4290.*] And in these reviews, many ion sources using nebulized feeds such as EESI [*Chem. Commun. 2006, 2042–2044.*], PSI [*Angew. Chem. 2010, 122, 889–892.*], DAI [*Anal. Chem. 2017, 89, 1059–1062.*], etc. are still classified as AMS. Based on this literature, we believe that the techniques described

in the manuscript remain within the “ambient mass spectrometry”. Understandably, different researchers have different opinions on the definition of AMS. Considering the Reviewer’s suggestion, we have rewritten the Introduction section and reduced the use of the words “AMS” in the text.

Rewrite section: “Ion sources are among the most essential components of mass spectrometry (MS) and have evolved from early electron impact (EI), chemical ionization (CI), and electrospray ionization (ESI) to various types of ambient ionization technologies, providing multidimensional application scenarios for mass spectrometry analysis. As mass spectrometry technology matures and becomes commercial, an increasing number of ionization techniques are being developed in open air due to their unprocessed or minimally pretreatment process, widely suitable to MS instruments, and good analytical performance¹. In the last 20 years, researchers have introduced and modified dozens of new ambient ionization technologies, which utilize diverse physicochemical processes, such as electrospray², laser ablation³, plasma⁴, thermal desorption⁵ and vibrational excitation⁶, and demonstrated impressive analytical performance in a variety of applications, most of which are defined as ambient mass spectrometry (AMS).

Due to the large variances in the chemical characteristics of the analytes, it is difficult to cover all the analytes by a single ionization technology. For example, ambient ionization technology based on an electrospray mechanism is usually utilized for nonvolatile polar compounds throughout an extensive mass range^{2,7,8}, and ionization techniques based on an atmospheric pressure chemical ionization (APCI) mechanism can characterize compounds that are nonpolar or exhibit a low polarity⁹⁻¹¹. An ion source with full analytical coverage is not only helpful for the analysis of complex samples, leading to the creation of more specific fingerprints, but also eliminates the need for additional mass spectrometry. For example, toxicology centers in advanced hospitals and toxicology laboratories usually contain three mass spectrometers (ESI, EI, and ICP-MS) to address different clinical situations involving poisoned patients. Therefore, the development of new ionization techniques to analyze various molecules with higher coverage is a major goal for many analytical chemists.”

Comment 3:

Ln 47-49: Here, the authors list the means that ambient MS is implemented to remove species from a sample surface, perhaps suggested (though poorly) in the phrase “to directly detect”.

Responds 3:

Thanks for your advice. We have re-written the phrase “to directly detect” according to the Reviewer’s suggestion. Change the phrase “to directly detect” to “demonstrate impressive analytical performance in a variety of applications”. The AMS techniques reported in the literature cited in Ln 47-49, some of the techniques in literature #4, along with literature #6, are not techniques for surface analysis. As stated in the previous responds, what we think of “AMS” is a more general concept.

Comment 4:

Ln 56: To which 3 mass spectrometers are the authors referring?

Responds 4:

Thanks for your advice. The three types of mass spectrometers we refer here are ESI (electrospray ionization)-MS, EI (electron impact)-MS and ICP (inductively coupled plasma)-MS. ESI is generally used in conjunction with LC for the analysis of polar and difficult to volatilize samples such as pesticides and psychotropic substances. EI-MS mainly focuses on the analysis of volatile organic compounds and benzene series. And ICP-MS is mainly used to analyze cases of heavy metal poisoning. Considering the Reviewer’s suggestion, we have added a description to the manuscript.

Comment 5:

Ln 81: The authors refer to the liquid sampling atmospheric pressure glow discharge source, ref 25. Indeed, this group has published a multitude of papers illustrating the concept of the combined atomic and molecular (CAM) ionization source. We believe that this is the concept to which the authors seek to address. The authors would do well to become familiar with this literature.

Responds 5:

Thanks for your advice. Once again, we apologize for the conceptual misleading of the word “digital”, the term “digital” as used in the text refers to the programmable form of the input power. In addition, we have read in detail about the excellent work of Marcus et al, and received inspiration. However, we find that Marcus's work has large discrepancies with the manuscript, mainly including: (a) liquid sampling atmospheric pressure glow discharge (LS-APGD) does not allow precise and fast regulation of the ionization energy, and the ratio of the peaks of molecular ions or proton ions is more determined by the solvent [*J. Anal. At. Spectrom.* 2020, DOI: 10.1039/d0ja00373e]. (b) LS-APGD seems to show low gas

temperatures in the reported literature [*J. Anal. At. Spectrom.* 2017, DOI: 10.1039/c7ja00008a], whereas our device allows for the regulation of gas temperatures in the ionization zone up to 1000 °C. Uncontrollable low temperatures result in lower desolvation capacity, the (c) ionization performance of LS-APGD is strongly influenced by the solvent, which is 2%-5% HNO₃ added to the solvent when treating elements [*Anal. Chem.* 2011, 83, 2425–2429]. (d) No reports support that LS-APGD can achieve controlled in-source cleavage, the ionization products of polar compounds are all protonated [*J. Anal. At. Spectrom.*, 2016, 31, 14]. (e) LS-APGD has not been reported to have the ability to detect organometallics and dissociate metal monomers from them. (f) The operating conditions of LS-APGD are greatly influenced by the flow rate and sheath gas [*Anal. Chem.* 2011, 83, 2425–2429], which are generally limited to 0.1 mL/min, which is a limitation when coupled with the LC. In our manuscript, the concept of the combined atomic and molecular (CAM) ionization source is not the only thing we seek to address. Ionization benefits from programmed control of power include, but are not limited to high coverage ionization, controlled in-source cleavage, wide range of controllable gas temperatures in the ionization range, analysis of organometallic forms, fast energy scanning for specific compounds, good liquid-phase coupling capability. Therefore, we believe that the ion source described in the manuscript has certain advantages in terms of functionality and analytical performance compared to the Marcus's work.

Comment 6:

Ln 91-94: Neither the previous reports of MIP or ICP sources “allow the dissociation of charged metallic elements”. This is phenomenologically incorrect and we again encourage the authors to be more knowledgeable in the literature of plasma source mass spectrometry.

Responds 6:

Thanks for your advice. We are very sorry for our incorrect writing “allow the dissociation of charged metallic elements”. In fact, the point we are trying to make is that MIP and ICP can directly detect metals in liquid electrolytes [*Anal Chem* 90, 13443-13450]. Considering the Reviewer's suggestion, we have changed the expression of this part of the sentence to read “allow the direct detection of metal elements in liquid sample”.

Comment 7:

Ln 102-103: We do not at all understand the term “atomization injection”. No form of nebulization (which is what is being done here) introduces free atoms. Indeed, throughout the

manuscript, including the Methods section, the description at times suggests an electrospray source and in other a pneumatic nebulizer. This lack of clarity is confusing and suggests a lack of understanding on the part of the authors.

Responds 7:

Thanks for your advice. We apologize for the misunderstanding caused by the term “atomization injection” and the unclear nebulization method in the manuscript. In fact, “atomization” here simply means nebulization, not introduces free atoms, the nebulized injection is also described as “atomization” in some literature [Science, 246, 4926, 64-71 (1989); Anal. Chem. 2017, 89, 1059–1062, Talanta, 255, 124237 (2023)]. To avoid ambiguity, we changed it to “pneumatic nebulization”. The revised sentence: “Liquid sampling experiment were carried out using pneumatic nebulization (quartz nebulizer with concentric structure), using N₂ as carrier gas (1000-3000 mL/min).”

Comment 8:

Ln 113: Any frequency and amplitude”? Surely there are some limitations.

Responds 8:

Thanks for your advice. We apologize for the lack of rigor in this sentence; in fact, we refer to the frequency limit of the input waveform in the subsequent sentence (Ln 116-118). Considering the Reviewer’s suggestion, we have added the limitations of frequency and amplitude in the text as: “modifiable frequency and amplitude (frequency \leq 10 Hz, amplitude within 0-2.5 V)”.

Comment 9:

Ln 128: The “intersection” of what and what? Here again, the regions of plasma structure and sampling are very well defined in the relevant plasma literature.

Responds 9:

Thanks for your advice. The term “intersection” here refers to the location where the plasma is focused. Since the microwave plasma torch essentially consists of rotating strips of plasma, an intersection point occurs, which was named plasma core in the early literature [*Spectrochim. Acta, Part B. 1991, 46B, 417-430.*]. Details of the naming of the various locations of the microwave plasma are shown in the original **Supplementary Fig. 1**.

Considering the Reviewer's suggestion, we decided to follow the nomenclature of the earlier literature and modify text and figure, here is the modified **Supplementary Fig. 1**.

Supplementary Fig. 1. Plasma morphology diagram. (a) Plasma length and shape are affected by different microwave power and carrier gas flow. (b) Plasma thermometry sites, including the first 5 mm of the tip, the plasma tip, the plasma core and the plasma root.

Comment 10:

Ln 129-131: What is meant by “temperature”? Is this the kinetic, ionic, electronic, vibrational, or excitation temperature? They each have a very specific meaning in the realm of plasma chemistry. How was it actually measured? The values presented have absolutely no meaning.

Responds 10:

Thanks for your advice. As stated in Ln 127, the meaning of temperature here is the gas temperature of the plasma. The gas temperature as a general characteristic varies significantly between different types of plasma [*ACS Energy Lett.* 2018, 3, 1013–1027]. We apologize for the lack of the gas temperature measurement method. We have added the measurement of gas temperature in the Methods section: “The temperature is measured by a thermocouple pyrometer (Mini Type K/J Input Thermometer, UNI-T Co. China) with a range of -50 to 1300°C. During the test, the temperature probe was placed at the target plasma position for 3 seconds, the gas temperature value was recorded and averaged over three repetitions.”

Comment 11:

Ln 149: We see no “schematic diagram” of the molecular ion peak . . .”.

Responds 11:

Thanks for your advice. We are very sorry for our negligence of the phrase “schematic diagram”. We decided to modify this figure note to read “comparison of molecular ion peak intensity and protonation peak intensity variation with microwave power.”

Comment 12:

All mass spectra of metal containing species: We note that in none of the presented spectra are their actual atomic ions, they appear as the metals with attached H, OH, etc. species. So, the “digital” situation is not realized. More importantly, while not readily apparent by the bare eye, magnification of the spectra reveals isotope abundances that are VERY MUCH in error. Indeed, if one were presenting a new ionization source for atomic species, some reporting of the determined isotopic abundances is required. The works of Marcus et al. in atomic mass spectrometry using Orbitrap analyzers would be very helpful to the authors

Responds 12:

Thanks for your advice. We have carefully studied the works of Marcus et al. Once again, we are sorry for the conceptual misleading of the word “digital”, the term “digital” as used in the text refers to the changing form of the input power. The manuscript does not mention that our aim is to detect atomic ions. For the question of isotope abundance, we have already placed the enlarged isotope mass spectra at the original **Supplementary Fig. 12** in the first submission of the manuscript, and the isotope abundances of tin are VERY MUCH correct as shown in the following figure. To further illustrate the accuracy of the isotopic abundances, we show the isotopic mass spectra of Pb and add it to the modified SI (**Supplementary Fig. 14**).

Supplementary Fig. 14. Isotope mass spectra of divalent lead, where m/z 208.99 is the highest abundance isotope ions $^{208}\text{PbH}^+$, m/z 207.98 is $^{207}\text{PbH}^+$ ions, m/z 206.98 is $^{206}\text{PbH}^+$ ions.

Comment 13:

Fig. 3 caption and discussion: The provided caption “WPMPI-MS provides metal analysis with in-source cleavage capability” is a complete misnomer. Only 2 of the compounds contain metals (again without any atomic ions observed), with the ApC and copper peptide not actually containing metals or indeed being dissociated to that level. Upon our looking on the internet, the term copper peptide is completely misused here. The presented molecule is a copper-binding peptide, but the authors present it as it contains Cu, which clearly is not evidenced in the spectra.

Responds 13:

Thanks for your advice. We are very sorry for our misnomer, and we have replaced the note to Fig. 3 with the following: “WPMPI-MS provides metal element analysis capability and biomolecular in-source cleavage capability.” to avoid misunderstanding. In fact, Fig. 3 is presented by combining two paragraphs of the figure for space reasons; (a)(b) in Fig. 3 corresponds to the paragraph for Ln 203, while (c)(d) corresponds to Ln 241, which led to a misunderstanding by the reviewers. What we are expressing is not the dissociation of metal atoms from ApC and copper peptide, we know very clearly that copper peptide does not contain copper, in fact it is a common tripeptide (Gly-His-Lys), and the use of ApC and

copper peptide is purely to demonstrate its ability to dissociate biomolecules. In order to avoid misunderstandings, we have replaced "copper peptide" with "Gly-His-Lys".

Comment 14:

Fig. 4a caption and discussion: a) What is/are superimposed mass spectra? If the different species are ionized at different applied powers, how is the data obtained? C) What is an "EIC diagram"? Are we looking at an extracted ion chromatogram (EIC)?

Responds 14:

Thanks for your advice. In response to the reviewer's query about Fig. 4a: It is because different species are ionized at different applied powers that we display the mass spectra with different applied powers superimposed on a uniform coordinate system and ultimately convert the vertical coordinate to relative abundance. As shown in the flow chart below, the top graph shows the total ion chromatogram graph after several power scans, with two points in a single scan demonstrating completely different mass spectra. Since their horizontal and vertical coordinates are harmonized, we can simply merge them together to obtain better displayed mass spectra (we can of course also analyze the mass spectra of a particular ion individually).

In response to the reviewer's query about Fig. 4c: Yes, we mentioned the full name (extracted ion chromatogram) of EIC in Ln 155; an EIC plot refers to a chromatogram at a specific mass-to-charge ratio, m/z is fixed to a particular ion, the horizontal coordinate is the retention time, and the vertical coordinate is the intensity (or relative abundance), which

better describes the spectra information of a particular compound. EIC plots are the most common way of characterizing data during chromatogram-mass spectrometry [*Anal. Chem.* 2021, 93, 36, 12181–12186.; *Environ. Sci. Technol.* 2019, 53, 550–559.; *Nature Communications* 2022, 13, 1347.], and have also been used to characterize specific ions in many mass spectrometry-related literatures [*Angew. Chem. Int. Ed.* 2022, 61, e202207098.; *J. Am. Chem. Soc.* 2015, 137, 7274–7277.; *Anal. Chem.* 2018, 90, 7154–7157].

Comment 15:

Ln 263-267: We are not told the method of computing the LOD or LOQ values for the various analytes. We have problems with response curves performed covering only 2 orders of magnitude in concentration space. If we look at the response curves in Fig. S17, the values for many of the compounds presented in Table S1 are not at all sensible. For example, looking at the curve for benzo[a]pyrene, we see that the curve ‘bottoms-out’ at approximately 150 µg/L, but the table reports an LOD of ~2 µg/L. This cannot be correct by standard means of calculation. Visually, very few of the curves would be expected to yield such low values as the table.

Responds 15:

Thanks for your advice. We are very sorry for our negligence of the method of computing the LOD or LOQ. The LOD or LOQ of this method was calculated using the following equation ($S/N=3, 10$.) based on measurements of spiked samples:

$$LOD = \frac{3c\sigma}{S}, \quad LOQ = \frac{10c\sigma}{S}$$

Where c is the analytes concentration in the spiked sample, σ is the standard deviation of all the measurements ($n=7$), S is the mean value of the signal intensity. This is a generalized way of calculating LOD and LOQ [*Anal. Chem.*, 2009, 81, 2426-2436.; *Chem. Commun.*, 2005, 1950-1952.; *J. Am. Chem. Soc.* 2019, 141, 72-75.], at the same time, this calculation method is independent of the results of the calibration curve. Considering the Reviewer’s suggestion, we have added the part to the Methods section of the manuscript.

Regarding the reviewer's query about the corrected curves, this is a result of the linear fitting method (weighting fitting), as our LOD and LOQ calculations are independent of the slope intercept of the curves, which will seem implausible when some of the fitted curves are compared to the LOD. However, this does not mean that our calibration curves and LODs are wrong. Considering the Reviewer’s suggestion, we used a direct fitting in plotting the calibration curves to minimize the linearity error at low concentrations and updated **Fig. S22** (original Fig. S17).

Supplementary Fig. 22. Linear equations for nine toxicological samples under serum extract matrices, the method demonstrated a good R^2 (0.9477-0.9996) over a linear range of $10 \mu\text{g/L}$ - $5000 \mu\text{g/L}$, where the limit of quantification for inorganic lead was slightly higher than for the remaining analytes.

Reviewer 2:

This manuscript is an exciting report of a new microwave plasma-based ionization source for mass spectrometry that is capable of rapidly modulating the power through application of digital waveforms. This research is an interesting advance in mass spectrometry ionization. This work presents numerous applications of this technology, supporting the claims of the authors. Building on this work could lead to an important new analytical tool in environmental speciation work, forensic science, and materials characterization. An ion source capable of both molecular and elemental analysis is compelling in a wide range of fields. It should be published after revisions. Revisions should primarily focus on providing more experimental details. Specific suggestions are listed below.

Comment 1:

I encourage the authors to be more precise in their language surrounding ambient ionization. Infusing samples in solution into an ion source is not generally considered ambient ionization. If the sample is not being introduced from a solid platform between the ion source and the MS inlet, it is not an ambient ionization experiment.

Responds 1:

Thanks for your advice. Considering the Reviewer's suggestion, we have supplemented our application experiments with direct detection of solid pharmaceutical powders to illustrate the analytical capabilities of WPMPI as an AMS.

In addition, the definition of the term "ambient mass spectrometry" as we understand it is more generally. As defined by Prof. R. Graham Cooks, ambient ionization refers to "the ionization of unprocessed or minimally modified samples in their native environment, and it typically refers to the ionization of condensed phase samples in air." [Ambient Mass Spectrometry. *Science* 2006, 311, 1566-1570.]. And in Facundo M. Fernandez's review, environmental mass spectrometry is defined as having the following characteristics: (a) ionization in the absence of enclosures such as those typically found in ESI. In other words, the technique should operate in the open air or ambient environment. (b) Ambient MS techniques allow direct ionization with minimum sample pretreatment such as preconcentration, extraction, derivatization, dissolution, or chromatographic or electrophoretic separation. Although this requirement can be relaxed to some extent in challenging applications of ambient MS. (c) Should be interfaceable to most types of mass spectrometers fit with differentially pumped atmospheric pressure interfaces, without substantial modification to the ion transfer optics or vacuum interface. (d) Should generate ions softly. [Ambient Sampling/Ionization Mass Spectrometry: Applications and Current Trends. *Anal. Chem.* 2011, 83, 4508–4538]. Over the past 15 years, innovations in the field of environmental ionization mass spectrometry have evolved rapidly, allowing these techniques to progressively outgrow their concepts. [Ambient Ionization Mass Spectrometry: Recent Developments and Applications. *Anal. Chem.* 2019, 91, 4266–4290]. And in these reviews, many ion source using nebulized feeds such as EESI [Chem. Commun. 2006, 2042–2044.], PSI [Angew. Chem. 2010, 122, 889–892.], DAI [Anal. Chem. 2017, 89, 1059–1062.], etc. are still classified as AMS.

Based on this literature, we believe that the techniques described in the manuscript remain within the "ambient mass spectrometry". Understandably, different researchers have different opinions on the definition of AMS. Taking into account the reviewers' suggestions, we have rewritten the Introduction section and modified the description in the manuscript by

reducing the use of the term "AMS" in the text and not defining the WPMPI-MS in the manuscript directly as an AMS technology.

Comment 2:

The specific design of the microwave plasma is critical to the data presented and is largely overlooked. It is briefly described in words as some kind of coaxial design, but the description is inadequate. While microwave plasmas are less fickle than inductively coupled plasmas, a specific configuration of a microwave plasma is generally only stable under a very limited range of power and gas conditions. The core of this paper is that the authors have managed to maintain a plasma over a range of energies at a constant gas flow while rapidly fluctuating the power. There must be some innovation in the design of this microwave plasma to accomplish these things. Please provide significant details. Is this a microwave-induced plasma as opposed to a capacitively coupled microwave plasma? Is this a non-resonant design? Did the authors encounter issues with impedance matching and reflected power over this range of plasma conditions? If this is based on a previous microwave plasma design, cite that work and describe the authors' modifications. Provide substantial details and figures/schematics to explain this innovative design at the heart of the exciting data presented in the manuscript.

Responds 2:

Thank you for your professional advice, we apologize for the overlook of the descript specific design of the microwave plasma. The microwave plasma is microwave induced, not capacitively coupled. It is a resonant design by setting a resonant cavity where microwave energy is transmitted into the resonant cavity through a standard N-type connector and forms a maximum electric field and excites the plasma at the end plane of the cavity. In order to realize the fast and stable switching of the microwave energy, firstly, the size of the resonant cavity is increased compared with the microwave plasma device used by the previous work [Rapid Commun Mass Spectrom. 2017; 31: 2092–2100, J. Mass Spectrom. 2013, 48, 669–676], the detailed diameter of the microwave plasma tube is now as follow: outer tube (44 mm O.D., 21 mm I.D.), central tube (7 mm O.D., 5 mm I.D.) and intermediate tube (3.0 mm O.D., 1.5 mm I.D.). The motion distance of the tuning end plane is 40 mm. The larger size results in better heat dissipation and the relatively larger power range supported, also increases the travel range for impedance matching adjustment. The choice of working gas, argon, is critical. Compared to helium, nitrogen or air, argon has the lowest discharge power.

In this design, the microwave power of about 40 W can activate the relatively stable plasma torch, which leads to an soft ionization zone with relatively low temperature and energy.

The adjustment of the resonant cavity is also crucial. After the microwave plasma works, the end plane in the rectangular tube needs to be adjusted. Real-time monitoring of reflected power by means of oscilloscopes in the microwave power supply to minimize the value of reflected power for impedance matching. The actual test results show that when the tuning end face and resonant cavity end face are 29 mm apart, the microwave reflection is minimal, resulting in a microwave energy utilization of more than 95%. In order to increase the stability of the microwave power supply's fast power switching operation, a circulator is added to the power supply output. The reflected power during plasma operation is guided to the forward output through the circulator, avoiding the direct impact of the reflected power on the microwave power supply so as to increase the stability of the power supply's forward output power. The following figure illustrates these design details, and we have added the relevant content in the **Methods Section** and in the **Supporting Information**.

Supplementary Fig. 26. The specific design drawing of the main part of WPMPI.

Modifications in **Methods Section**: “Liquid sampling experiment were carried out using pneumatic nebulization (quartz nebulizer with concentric structure), using N_2 as carrier gas (1000-3000 mL/min). The sample solution is injected into the quartz nebulizer by a syringe pump. The microwave plasma generator is a resonant cavity design consists of microwave outer tube, central tube, and intermediate tube, where the microwave energy forms a maximum electric field and excites the plasma at the end plane of the cavity. In order to achieve fast and stable switching of microwave energy, we have optimized the design and impedance matching of the device details, a circulator is added to the output of the power supply, the details are

shown in the Supplementary Information, and the microwave energy utilization reaches over 95%.”

Comment 3:

While the microwave plasma design is the most critical detail, the authors should also elaborate on other experimental details. Specific instances are listed below:

- How was the temperature of the plasma measured? How were the reagent gas ions in the plasma measured? These smaller ions can be difficult to monitor in an LTQ-Orbitrap. Did the authors measure these ions directly with this instrument? Did they use an optical method?

Responds 3:

Thanks for your advice. We are very sorry for the negligence of other experimental details. The temperature is measured by a thermocouple pyrometer (Mini Type K/J Input Thermometer, UNI-T Co. China) with a range of -50 to 1300°C. During the test, the temperature probe was placed at the target plasma position for 3 seconds, the gas temperature value was recorded and averaged over three repetitions.

The reagent gas ions detected by a home-made time-of-flight mass spectrometry (Assisted development by Suzhou Zhipu Weixing Intelligent Technology Co.). Tests for reagent ions have been studied in our previous work, and we have added references in manuscript and added specific instruments and methods in the **Methods Section**: “The temperature is measured by a thermocouple pyrometer (Mini Type K/J Input Thermometer, UNI-T Co. China) with a range of -50 to 1300°C. During the test, the temperature probe was placed at the target plasma position for 3 seconds, the gas temperature value was recorded and averaged over three repetitions.”. and “The reagent gas ions detected by a home-made time-of-flight mass spectrometry (Assisted development by Suzhou Zhipu Weixing Intelligent Technology Co.).”

Comment 4:

Did the populations of reagent ions shift in relative abundance with shifting power? This is an important detail, given the observed shifts from protonated molecules to molecular ions that were observed in the spectra with increases in applied power. Presumably, the population of species more likely to cause penning ionization, like oxygen radicals, would increase with increasing power, while the hydronium ions more likely to promote proton transfer would decrease.

Responds 4:

Thank you for your advice. This is a very professional and interesting question. According to the Reviewer's suggestion, we conducted additional experiments, as shown in the figure below, the populations of reagent ions shift in relative abundance with shifting power. It is evident that the hydronium ions ($(\text{H}_2\text{O})_n\text{H}^+$), which cause proton transfer, decrease significantly with increasing power; while reagent ion species that can bring molecular ions, such as oxygen radicals and nitrogen radical ions, increase in abundance. We sincerely appreciate your professional advice and have added this part to the **manuscript**, which we believe will help to improve the quality of the article: "To some extent the phenomenon may be related to the reagent gas ion species, as we have noted that the populations of reagent ions shift in relative abundance with shifting power (Fig. S3). It is evident that the hydronium ions ($(\text{H}_2\text{O})_n\text{H}^+$), which cause proton transfer, decrease significantly with increasing power; while reagent ion species more likely to cause penning ionization that can bring molecular ions, such as oxygen radicals and nitrogen radical ions, increase in abundance." The reagent gas ions were detected by a home-made time-of-flight mass spectrometry (Assisted development by Suzhou Zhipu Weixing Intelligent Technology Co.).

Supplementary Fig. 3. The populations of reagent ions shift in relative abundance with shifting power. Hydronium ions ($(\text{H}_2\text{O})_n\text{H}^+$ green & blue), which cause proton transfer, decrease significantly with increasing power; while reagent ion species that can bring molecular ions, such as oxygen radicals (purple) nitrogen radical ions (yellow), increase in abundance.

Comment 5:

For the nebulizer used to introduce the sample into the plasma, the authors note that they can introduce 0-5 kV to it. This is a critical detail. If the authors are applying high voltage to

the nebulizer, they are doing ESI in addition to any microwave plasma. We would expect many of the spectra presented to appear if only ESI was used (none of the spectra with EI-like fragmentation). The authors should carefully detail when and what voltages were applied to the nebulizer and how that voltage was coupled to it. They should analyze how the voltage applied at the nebulizer influenced the spectra and provide data to support their conclusions.

Responds 5:

Thank you for your advice. We are very sorry for the unclear presentation of the nebulizing voltage. In fact, pneumatic nebulizer was used for all experiments in this manuscript, no additional voltage is applied to the system, except where stated. In the Methods section, we mentioned that electrospray voltage could be **selected** (Just to illustrate that the nebulization mode is not limited to a pneumatic nebulizer), this seems to have created misunderstandings. Therefore, we replaced the term “atomization injection” with " pneumatic nebulizer" and removed the contents of “0-5 kV” in the Methods section.

Comment 6:

Concentrations of samples should be listed for all the data presented. It is provided for some experiments and not for others. Concentrations are explicitly listed for the serum studies but not for the other example analytes listed prior to those experiments.

Responds 6:

Thank you for your advice. We are very sorry for the oversight in the concentration descriptions of some experiments. We add details of the concentrations of those experiments in the manuscript: “For the preparation of standards, PAHs were dissolved in acetonitrile and the rest were dissolved in water : methanol (1:1). In the standard test experiments, as well as in-source cleavage experiments, a concentration of 10 mg/L was used for all samples.”

Comment 7:

Figure S5 shows a mixture of PAHs ionized at moderate powers. There is not significant fragmentation observed in this spectrum. In Figures S6 and S7, there is significant fragmentation. What has changed? Presumably, this is due to an increase in power, but it is quite important that the authors specify this. I would encourage the authors to include the power applied and sample concentration in each of the captions for the spectra in the main body and supplementary information.

Responds 7:

Thank you for your advice. In original Fig. S5, we show four classical PAHs without heteroatoms, and their structures are relatively stable, as shown in figure below; whereas the compounds used in original Fig. S6, and S7 are anthraquinone and ethyl acetate, their structures are more fragile compared to the compounds in Fig. S5, and this section is investigated for the elimination and rearrangement reactions occurring for the dissociation of certain compounds in WPMPI.

Of course, as you suggested that the increase in power has a part of the effect, and taking into account the reviewer's comments, we have added the corresponding power values in each figure including Fig. 3; Fig. S5, S6, S7, S8, S9, S10, S11, S16.

Comment 8:

The authors note that during operation of the plasma, a metal tip is applied to the intermediate tube. Is this tip grounded? What is it connected to? Is it applied throughout the experiment, or only in the beginning? What is it composed of, and is any metal contamination from this tip observed in the spectra at high powers?

Responds 8:

Thank you for your advice. The purpose of the metal tip is to provide the electrons that activate the plasma (similar to an ignition device), this tip does not need to be grounded, and no specific material requirements, a normal syringe tip will suffice. This triggering process is only applied in the beginning and is removed immediately after plasma activation, thus not leading to contamination of the spectrum. Considering the Reviewer's suggestion, we have added description in the **Methods Section**: "During operation, the plasma needs to be excited by touching the intermediate tube with a metal tip (This process is only applied in the beginning and is removed immediately after plasma activation, the tip does not need to be grounded, and no specific material requirements)."

Comment 9:

I encourage the authors to superimpose the waveforms on the EIC data in Figures 2 and S3. The correlated timing of the applied power and MS signal is an interesting detail that should be commented on.

Responds 9:

Thank you for your advice. We followed the reviewer's comments and superimpose the waveforms on the EIC data in Fig. S4 (original Fig. S3):

Supplementary Fig. 4. EIC plots of reserpine and divalent lead when ionized with different scanning waveforms (100-200 W, red line) (a) EIC plots for both ions when sawtooth wave at 0.2 Hz is applied, the blue line represents the EIC plot of divalent lead ions within 0.2 min, and the black is the EIC plot of reserpine. (b) EIC plots when triangle wave at 0.2 Hz is applied, the orange line represents the EIC plot of divalent lead ions within 0.2 min, and the black is the EIC plot of reserpine.

Reviewer 3:

In this study, a digitally regulating microwave plasma energy device (named wide-energy programmable microwave plasma ionization, WPMPI) was developed and applied to ionize analytes introduced through nebulizing or electrospraying liquid samples. Different ion types were detected by the WPMPI-MS, including radical ions for PAHs, protonated ions for

organic standards and multiply charged ions for large biochemical compounds. Unfortunately, the development of microwave plasma ionization technology (references #27-30 and more) as well as changing the high voltage applied to gas molecules to vary the plasma energy to induce different ionization mechanisms (reference #15 and more) have previously been reported, meaning the novelty of this study is not high. In addition, several substantial inquiries as listed below mean this reviewer is overall unable to support the publication of this manuscript in Nature Communication.

Comment 1:

This manuscript does not adequately explain how the developed method is novel; specifically, the significance of the formation of multiply charged analyte ions in the WPMPI (P.9, line 173-174 and Fig. S4c) is not explained upon by the authors. During WPMPI-MS analysis, sample solution was introduced into the source through nebulization or ESI mechanisms (p.17, line 324-327). Therefore, WPMPI seems to be a combination of two ionization methods: ESI and plasma APCI. But the concept of dual ionization source has previously been reported in several publications, where plasma or corona discharge APCI was used to ionize low-to-medium polarity compounds (to form radical or protonated analyte ions) and ESI was used to ionize high polarity compounds (to form protonated or multiply charged analyte ions). The plasma energy could be changed simply by varying the voltage applied to gas molecules to induce different ionization mechanisms, with one of the examples being reference #15 in this manuscript.

Responds 1:

Thank you for your advice. The innovation of our proposed method focuses on the use of waveform programming for rapid scanning over a wide energy range, which enables various analytical functions (e.g. simultaneous detection of heavy metal elements and drug molecules in less than a second). For the multiply charged analyte ions in Fig. S5, we believe that this is due to the action of thermal spray (plasma heat the spray plume) and hydronium ions, which has been reported in some literature (Anal. Chem. 2018, 90, 7239-7245. Analyst, 2021, 146, 5682-5690). In addition, we are very sorry for the unclear presentation of the nebulizing voltage. In fact, pneumatic nebulizer was used for all experiments in this manuscript, no additional high voltage is applied to the system. In the Methods section, we mentioned that electrospray voltage could be **selected** (Just to illustrate that the nebulization mode is not limited to a pneumatic nebulizer), this seems to have created misunderstandings. Therefore, we replaced the term "atomization injection" with " pneumatic nebulizer" and removed the

contents of “0-5 kV” in the Methods Section. In WPMPI, the nebulizer is only used as an injection method, due to liquid-phase injection being adaptable to a wider range of application scenarios and it is still a single ion source. In contrast to other literature, such as reference #15, which discusses the effects of varying the voltage to induce different ionization mechanisms, the energy change induced by altering the voltage of a general plasma source (such as DBD) is much weaker compared to WPMPI. General plasma ion source such as APCI or DBDI does not have a controllable gas temperature, not capable of elemental analysis, not able to adjust the ionization energy quickly, and it is less effective in ionizing polar compounds. In addition, we complemented the relevant experiments by validating the ability of the method to directly analyze solid samples.

Supplementary Fig. 23. WPMPI-MS mass spectra of the solid sample with complex matrices: Yunnan Baiyao powder (200 µg) in 100-140 W.

Supplementary table 2. Fourteen high abundant ion signals were screened out and preliminary identified as representative diterpenoids in aconitum.

No.	Compounds	Measured m/z	Exact m/z	Mass error ppm (n=5)	Formula	Ref
1	Hetisinone	328.1914	328.1908	1.83	C ₂₀ H ₂₅ NO ₃	2
2	Actaline	342.2432	342.2428	1.17	C ₂₂ H ₃₁ NO ₂	3
3	Heterophyllinine-A	344.2588	344.2583	1.45	C ₂₂ H ₃₃ NO ₂	4
4	Atidine	360.2537	360.2534	0.83	C ₂₂ H ₃₃ NO ₃	2
5	Corumdizine	374.2700	374.2691	2.40	C ₂₃ H ₃₅ NO ₃	3
6	Daphnioldhanine K	388.2501	388.2483	4.63	C ₂₃ H ₃₃ NO ₄	5
7	Aconasutine	404.2805	404.2796	2.23	C ₂₄ H ₃₇ NO ₄	6
8	14-Dehydrotalatisamine	420.2754	420.2745	2.14	C ₂₄ H ₃₇ NO ₅	7
9	Jadwarine-B	434.2921	434.2901	4.61	C ₂₅ H ₃₉ NO ₅	8
10	Vaginaline	452.2682	452.2643	8.62	C ₂₄ H ₃₇ NO ₇	9
11	Pyrochasmaconitine	554.3127	554.3113	2.52	C ₃₂ H ₄₃ NO ₇	10

12	Mithaconitine	570.3072	570.3062	1.75	C ₃₂ H ₄₃ NO ₈	11
13	Sinonapelloinine B	602.3340	602.3324	2.66	C ₃₃ H ₄₇ NO ₉	12
14	8-Deacetylyunaconitine	618.3289	618.3273	2.59	C ₃₃ H ₄₇ NO ₁₀	13

Thus, it is to illustrate that spraying is only an injection method and that WPMPI is not a dual ion source. We have added the Figure and corresponding text to the SI and the manuscript, which we hope will help to improve the manuscript.

Comment 2:

As is generally known, different high energy species are present in plasma including radicals, protons, electrons, and cluster ions. These species can react with the analytes to form different types of analyte ions. It is then not surprising to see radical and protonated analyte ions on plasma-based WPMPI mass spectra. However, the study would greatly benefit from presenting and comparing mass spectra obtained from EI (in vacuum) and WPMPI (ambient conditions) to better consider the similarities and differences between the two ionization methods.

Responds 2:

Thank you for your advice. We fully agree with the reviewer's suggestion that it is very common to see protonated or radical analyte ions in plasma ion sources. However, the method described in the manuscript enables the modulation of the abundance of protonated or radical ions, including some of the primary reagent ions, by rapidly varying the programmed plasma energy. As shown in the figure below, it is evident that the hydronium ions ((H₂O)_nH⁺), which cause proton transfer, decrease significantly with increasing power, while reagent ion species that can form molecular ions, such as oxygen radicals and nitrogen radical ions, increase in abundance. This process is rarely reported in the literature on other plasma ion sources.

Supplementary Fig. 3. The populations of reagent ions shift in relative abundance with shifting power. Hydronium ions ((H₂O)_nH⁺ green & blue), which cause proton transfer, decrease significantly with increasing power; while reagent ion species that can bring molecular ions, such as oxygen radicals (purple) nitrogen radical ions (yellow), increase in abundance.

In addition, WPMPI exhibits similar ionization effects to EI within a certain energy range. For example, the spectrum of WPMPI for PAHs at the 140-180 W energy gradient is similar as that of EI (original Figure S5), and the fragmentation pattern of WPMPI also shares some similarities with EI (original Figure S6, 7). However, the ionization ability of WPMPI for metal elements and non-volatile substances is difficult to be achieved by EI. Considering the Reviewer's suggestion, we have added a comparison of WPMPI with various common ion sources to illustrate the advantages.

Table 1. Comparison of analyte types between WPMPI and conventional ion sources

Analyte type	EI	ESI	ICP	APCI	MALDI	WPMPI
Polar	○	○	×	□	○	○
Non-polar	○	×	×	○	○	○
Volatile	○	○	×	○	×	○
Non-volatile	×	○	×	□	○	○
Metal element	×	×	○	×	×	○
In-source cleavage	○	×	×	□	□	○

* ○ Detectable; × Undetectable; □ Depending on the situation.

We have added the charts and corresponding text to the SI and the manuscript, which we hope will help to improve the manuscript.

Comment 3:

Using microwaves in ambient ionization sources is a major safety concern. Exposure to microwaves can cause mild skin burns or cataracts and a piece of metal can act as an antenna, which can create sparks and start a fire.

Responds 3:

Thank you for your advice. We understand reviewers' concerns about security, in fact, we designed it with that in mind. Firstly, we applied perfect shielding to both the power module and the transmission line, and used professional RF connecting lines to place microwave leakage (**Supplementary Fig. 24a**). Subsequently, we tested the microwave radiation (HT-M2 Microwave Leakage Detector, Dongguan Xintai Instrument Co., Ltd.) in the working condition of the device, and the results are shown in **Supplementary Fig. 24b-d**. In

the main part of the rectangular tube, even if the detector is close to the surface, the measured microwave radiation value is only about 1.21 mW/cm^2 , making it nearly undetectable beyond 10 cm. In the front section of the plasma, the microwave leakage at 3 cm is approximately 3.38 mW/cm^2 , which is lower than the standards of various countries (Chinese National Standard 5 mW/cm^2 , and United States Standard 10 mW/cm^2 , operating position), and the attenuation outside 10 cm is 0.78 mW/cm^2 , which is hardly detectable outside 20 cm. In addition, during general operation, the operator does not need to be within 20 cm of the front. Therefore, we are confident that the operator's safety can be ensured. Additions in manuscript and SI: “In addition, we discussed the safety of using microwaves in ambient ionization sources (**Fig. S24**). Microwave leakage is as low as 0.78 mW/cm^2 at 10 cm of the plasma tube, which is undetectable from 20 centimeters away, allowing the safety of the operator to be guaranteed.”

Supplementary Fig. 24. The microwave radiation levels were measured at various locations during the operation of WPMPI: a) connecting lines, b) stick close to the main body, c) 3 cm from the plasma tip, d). 10 cm from the plasma tip.

Comment 4:

The concentrations of the standard sample solutions used in this study were not reported at all (see text and Figs. 1-3 and Figs. S3-S15).

Responds 4:

Thank you for your advice. We are very sorry for the oversight in the concentration descriptions of some experiments. We add details of the concentrations of those experiments in the manuscript: “For the preparation of standards, PAHs were dissolved in acetonitrile and the

rest were dissolved in water : methanol (1:1). In the standard test experiments, as well as in-source cleavage experiments, a concentration of 10 mg/L was used for all samples.”

Comment 5:

The authors claim that the ion signals of 9 standards spiked in the serum were detected by WPMP-MS simply after protein precipitation and with no serious interferences from blood matrices (P. 13, line 255-257 and Fig. 4a). This would not be a logical conclusion unless the concentration of the spiked standards is high. Indeed, my own calculations place the sample concentration to be at a high level - 10 ppm (P.17, line 312-316), thus explaining why the ion signals of spiked standards were detected.

Responds 5:

Thank you for your advice. Yes, the stronger ionic signal shown in Fig. 4a was measured when spiking serum at a concentration of 10 ppm. This experiment was performed to validate the concept of the method for energy scanning of multiple compounds with widely varying properties. The use of higher concentrations of spiked samples resulted in clearer and more aesthetically pleasing spectra. In subsequent experiments, we also tested spiked samples with concentrations as low as 10 ppb and still found the target ions of the compounds, as displayed in the figure below. All data were validated by high-resolution mass spectrometry and checked against blanks to ensure the reliability of the results.

Comment 6:

The authors claim that the sensitivity for the detection of spiked standards in serum with WPMPI-MS is at sub-ppt level (P. 13, line 264-265). Since the sensitivity of most ambient ionization mass spectrometric technologies is at the ppb level, it would be very difficult to achieve such high detection sensitivities from complex serum samples without additional sample processes like extraction, concentration, and chromatographic separation.

Responds 6:

Thank you for your advice. We understand your concern. In fact, we reported the LODs in the range of 0.88-14.49 $\mu\text{g/L}$ in the original manuscript (line 264), which was at ppb level. In most of the samples we tested, the limit of detection (LOD) was maintained at the ppb level, and only two samples had LODs of 880, 980 ppt (sub-ppb). Besides, we added a 4-fold amount of acetonitrile as a solvent to precipitate the proteins during the analyses (line. 317), which greatly reduced the concentration of the background substrate.

Reviewer 4:

Authors present an ionization technique that is based on the microwave plasma. The work is based on previously known technology that is applied in a new way. Authors are aiming to digitalize the ionization and also broaden the range of analytes.

In general, the results shown here are promising and could be of great value in the origin of life field.

Comment 1:

Since the paper is not only a proof of concept anymore and is targeted to show that the technology is applicable for wide range of analytes, the paper is lacking this variability of analytes. Analyzing 9 toxicological standards is enough for proof of concept but for this paper application for wider range of analytes should be shown.

Responds 1: Thank you for your advice. We expanded our study to include two additional application experiments: testing of spiked soil extractants and direct detection of solid pharmaceuticals, demonstrating the universality of the WPMPI in analyzing various types of analytes under complex matrices. The details are as follows:

Three perfluoroalkyl compounds (perfluorobutane sulfonic acid (PFBS), potassium perfluoro-1-octanesulfonate (PFOS-K), Pentadecafluorooctanoic acid (PFOA)) and cadmium stearate were added to the soil extraction solution to formulate a mixture of 1 mg/L. The samples were analyzed by WPMPPI using a stepped wave of 80-200 W (< 30 s). Four samples were measured at different power steps (Fig. 5): PFOA, which is prone to cleavage, showed the highest signal intensity at 80 W and could not be seen above 120 W; PFOS-K and PFBS are similar in chemical properties and more stable, had an optimal ionization power of 160W; and cadmium metal, which needed to raise the power to 200 W before a clear isotopic signal could be observed.

Fig. 5. Detection of spiked soil extracts using WPMPPI stepped wave power scan mode. (a) WPMPPI mass spectra of PFBS and PFOS-k at the step of 160 W. (b) WPMPPI mass spectra of PFOA at the step of 80 W. (c) WPMPPI mass spectra of cadmium at the step of 200 W.

In order to demonstrate the certain anti-matrix interference capability of WPMPPI, we conducted WPMPPI-MS analysis on diterpenoid alkaloids in Yunnan Baiyao powder under 100 W conditions. It only requires dipping a few micrograms of powder sample with a quartz capillary to generate stable and persistent mass spectrometry signals. Protonated diterpenoid alkaloids were clearly observed within the mass range of m/z 300-700. Fourteen high abundant ion signals were selected, and preliminary identification was performed using high-

resolution mass spectrometry, revealing them as representative diterpenoid compounds extracted from Aconitum.

Supplementary Fig. 23. WPMPI-MS mass spectra of the solid sample with complex matrices: Yunnan Baiyao powder (200 μ g) in 100-140 W.

Supplementary table 2. Fourteen high abundant ion signals were screened out and preliminary identified as representative diterpenoids in aconitum.

No.	Compounds	Measured m/z	Exact m/z	Mass error ppm (n=5)	Formula	Ref
1	Hetisinone	328.1914	328.1908	1.83	C ₂₀ H ₂₅ NO ₃	2
2	Actaline	342.2432	342.2428	1.17	C ₂₂ H ₃₁ NO ₂	3
3	Heterophyllinine-A	344.2588	344.2583	1.45	C ₂₂ H ₃₃ NO ₂	4
4	Atidine	360.2537	360.2534	0.83	C ₂₂ H ₃₃ NO ₃	2
5	Corumdizine	374.2700	374.2691	2.40	C ₂₃ H ₃₅ NO ₃	3
6	Daphnioldhanine K	388.2501	388.2483	4.63	C ₂₃ H ₃₃ NO ₄	5
7	Aconasutine	404.2805	404.2796	2.23	C ₂₄ H ₃₇ NO ₄	6
8	14-Dehydrotalatisamine	420.2754	420.2745	2.14	C ₂₄ H ₃₇ NO ₅	7
9	Jadwarine-B	434.2921	434.2901	4.61	C ₂₅ H ₃₉ NO ₅	8
10	Vaginaline	452.2682	452.2643	8.62	C ₂₄ H ₃₇ NO ₇	9
11	Pyrochasmaconitine	554.3127	554.3113	2.52	C ₃₂ H ₄₃ NO ₇	10
12	Mithaconitine	570.3072	570.3062	1.75	C ₃₂ H ₄₃ NO ₈	11
13	Sinonapelloinine B	602.3340	602.3324	2.66	C ₃₃ H ₄₇ NO ₉	12
14	8-Deacetylyunaconitine	618.3289	618.3273	2.59	C ₃₃ H ₄₇ NO ₁₀	13

We have added the experiments and results to the manuscript and SI, which we hope will contribute to the quality of the manuscript.

Comment 2:

One of the main drawbacks of the paper is that it is missing characterization of the ionization. For example, it lacks discussion on the reproducibility of the signal or fragmentation patterns. Investigation of matrix effects would also provide additional information.

Responds 2:

Thank you for your advice. Your comments are very valuable and we apologize for the lack of ionization characterization. With reference to your suggestion, we have added stability experiments for the fragmentation patterns. We selected two compounds and tested their fragmentation ratios at four different power steps, recording them every week for a cumulative total of 28 days and calculating day-to-day RSD. The results are shown in **Supplementary Fig. 21**, the vertical coordinate is the fragment ion to parent ion intensity ratio, and the day-to-day RSD of the four fragment ion ratios ranged from 5.2 to 13.5 %. On the other hand, we mention in the manuscript the detection stability of the nine spiked samples under complex matrix conditions (serum), and Table S2 demonstrates that their intraday RSDs are in the range of 6.2-16.6%, which we consider tolerable for an open plasma ion source [Mass Spec Rev. 2021;1–36.].

Supplementary Fig. 21. Intensity ratios and of the highest abundance fragment ions of the two compounds at the four energy steps with WPMPI-MS.

In order to demonstrate the certain anti-matrix interference capability of WPMPI, we add two experiments for applications in complex matrices, described in the previous response.

We have added the experiments and results to the manuscript and SI, which we hope will contribute to the quality of the manuscript.

Comment 3:

Since the paper is claiming “unprecedented coverage” compared to conventional ion sources, a better comparison should be presented.

Responds 3:

Thank you for your advice. Considering your suggestion, we have added a comparison of our method with a variety of conventional ion sources, which is presented in a table as follows.

Table 1. Comparison of analyte types between WPMPI and conventional ion sources

Analyte type	EI	ESI	ICP	APCI	MALDI	WPMPI
Polar	○	○	×	□	○	○
Non-polar	○	×	×	○	○	○
Volatile	○	○	×	○	×	○
Non-volatile	×	○	×	□	○	○
Metal element	×	×	○	×	×	○
In-source cleavage	○	×	×	□	□	○

* ○ Detectable; × Undetectable; □ Depending on the situation.

We have added the charts and corresponding text to the SI and the manuscript, which we hope will help to improve the manuscript.

Comment 4:

Title refers to “digital ionization”. It is unclear what makes this ionization digital on comparison to previous techniques. Reviewer is in the opinion that by using programming for the ionization does not make the ionization digital.

Responds 4:

Thanks for your advice. We are very sorry for our negligence of the conceptual misleading of the word “digital” in this paper. The term "digital" used in the text refers to the changing form of the input power, which we programmatically split into a number (or two) of

energy levels that are relatively independent and can jump rapidly, thus enabling a fast ionization process with good coverage of different compounds. In this model, whether or not to ionize a compound can be modulated to some extent, as shown in Fig. 2 and Fig. 5 in the manuscript and Fig. 5, this type of ionization that we consider can be called "digital". Similar to the concept of "digital ion trap", digital waveform is introduced into the ion trap electrodes, instantaneously switching the frequency and duty cycle of the rectangular waveform to excite the ions instead of converting the amplitude [Int. J. Mass Spectrom. 221 (2002), 117–138. Anal. Chem. 78 (2006), 1995-2000]. So we changed the title to read: "Toward digital programmable ionization: Wide-energy programmable microwave plasma for high coverage mass spectrometry analysis of various compounds"

Comment 5:

Furthermore, it remains unclear what are the "trace biosamples" mentioned in the title of the paper.

Responds 5:

Thanks for your advice. We used a drop of blood as a sample in our application experiments, approximately 200 uL, for spiking tests. This magnitude is much smaller than the amount required for general biochemical analyses, so we use the "trace biosamples" in the title. During the review process, we added other application experiments and therefore revised the title to: "Toward digital programmable ionization: Wide-energy programmable microwave plasma for high coverage mass spectrometry analysis of various compounds"

Comment 6:

Work presents LoD and LoQ values but it is not specified how these are calculated.

Responds 6:

Thanks for your advice. We are very sorry for our negligence of the method of computing the LOD or LOQ. The LOD or LOQ of this method was calculated using the following equation (S/N=3, 10,) based on measurements of spiked samples:

$$LOD = \frac{3c\sigma}{S}, \quad LOQ = \frac{10c\sigma}{S}$$

Where c is the analytes concentration in the spiked sample, σ is the standard deviation of all the measurements (n=7), S is the mean value of the signal intensity. This is a generalized way of calculating LOD and LOQ [Anal. Chem., 2009, 81, 2426-2436.; Chem. Commun.,

2005, 1950-1952.; J. Am. Chem. Soc. 2019, 141, 72-75.], at the same time, this calculation method is independent of the results of the calibration curve. Considering your suggestion, we have added the part to the **Methods Section** of the manuscript.

Comment 7:

Line 215 - exhibits -> exhibit

Responds 7:

Thanks for your advice. Thank you for pointing out the grammatical error, we have corrected in the manuscript.

Comment 8:

The overall readability of the paper should be improved, especially sections related to results and discussion.

Responds 8:

Thanks for your advice. We carefully modified the results and discussion sections, consulted with native speakers, and revised a great number of statements to improve the readability of the article.

Comment 9:

The time period of the analysis has not been specified.

Responds 9:

Thanks for your advice. We calculated the time period of the analysis and added it to the **Methods Section** of the manuscript: "Injecting the sample through the nebulizer, one analysis time period is less than 3 minutes; when coupled with LC, it depends on the chromatography process. When analyzing solid samples (pharmaceutical powder), use a quartz capillary tube to dip a few micrograms and place it directly in the plasma area, the analysis time period is less than 30s."

Reviewer 5:

A wide-energy programmable microwave plasma ionization mass spectrometry (WPMPI-MS) is presented with various characteristics by digitally regulating the microwave energy. This is an interesting modification of the ion source and deserves further investigation. However, I think more work needs to be done before the paper can be published.

Comment 1:

In the WPMPI-MS system, the power of the microwave plasma (0-200 W) can be regulated by a ref voltage. The overall range in temperature variation covers 350 K-1300 K. It is recommended to increase the investigation of thermally unstable samples. The influence of excess energy produced and active species on the analysis of oxidizes easily compounds should be discussed.

Responds 1:

Thanks for your advice. This is an interesting question, although the WPMPI-MS has gas temperatures between 350K-1300K in the 60-200W power range, the device can be set to remain at 60W during use, at which point the temperature will be no more than 450K, compared to 500K-600K for the conventional mass spectrometry inlet ion transfer tubes, therefore we believe that the WPMPI has a certain ability to analyze thermally unstable compounds. On this basis, we added experiments on thermally unstable compounds and oxidizes easily compounds. Artemisinin is an important drug for the treatment of malaria, which is thermally unstable because of its dioxygen structure. We employed WPMPI-MS with 60 W energy to analyze the artemisinin solution. As shown in Fig. S6a below, abundant protonated molecular ions were observed (40%), indicating that the structure of artemisinin was preserved to a certain extent. Fig. S6b shows the mass spectrum of the oxidizes easily compound N-isopropyl-N'-phenyl-1,4-phenylenediamine (IPPD) when ionized using WPMIP at 100 W. It can be observed that the abundance of oxidation products accounts for only 9% and 4% (could be much lower at low power), with the main ionized product being the protonated molecular ion. Ascorbic acid (vitamin C) was also analyzed using WPMPI at 60 W, and surprisingly no significant oxidation products were observed (Fig. S6c), with the highest abundance of ionization products being deprotonated ions. Overall, we concluded that WPMPI can be applied to the analysis of most thermally unstable and oxidizes easily compounds. We have added this content to the manuscript and the SI, which we hope will help to improve the quality of the manuscript.

Additions to the manuscript: “In addition, maintaining the WPMPI at a low power (60W) enabled the analysis of some thermally unstable compounds and easily oxidizes compounds. Artemisinin is an important drug for the treatment of malaria, which is thermally unstable because of its dioxygen structure. We employed WPMPI-MS with 60 W energy to analyze the artemisinin solution, abundant protonated molecular ions were observed (40%), indicating that the structure of artemisinin was preserved to a certain extent (**Fig. S6a**). The easily oxidizable compound N-isopropyl-N'-phenyl-1,4-phenylenediamine (IPPD) was ionized using WPMIP at 100 W, resulting in an abundance of the oxidized products of only 9% and 4%, and the main ionization product was the protonated molecular ion (**Fig. S6b**). Ascorbic acid (vitamin C) was also analyzed using WPMPI at 60 W, and surprisingly no significant oxidation products were observed (**Fig. S6c**), with the highest abundance of ionization products being deprotonated ions.”

Supplementary Fig. 6. (a) WPMPI-MS with 60 W energy to analyze the artemisinin solution, abundant protonated molecular ions were observed (40%). (b) The easily oxidizable compound N-isopropyl-N'-phenyl-1,4-phenylenediamine (IPPD) ionized using WPMIP at 100 W. (c) The easily oxidizable compound ascorbic acid ionized using WPMIP at 60 W.

Comment 2:

A conclusion given by the author is this method can achieve high coverage of substances with various characteristics by digitally regulating the microwave energy. It should be increased whether the coverage of organic compounds is increased, especially whether the response of compounds that are not easily ionized by ESI is increased.

Responds 2:

Thank you for your advice. In the second part of the RESULT, we show the analytical results of PAHs compounds with WPMPI, none of which are ionizable by the ESI source [J Am Soc Mass Spectrom 23(2012), 530-536.]. Considering your suggestion, we have added a comparison of our method with a variety of conventional ion sources, which is presented in a table as follows. In addition, we conducted experiments to detect ferrocene and methyl phenyl sulfide using WPMPI. These samples could not be detected by ESI, but molecular ion peaks were observed when using WPMPI-MS.

Table 1. Comparison of analyte types between WPMPI and conventional ion sources

Analyte type	EI	ESI	ICP	APCI	MALDI	WPMPI
Polar	○	○	×	□	○	○
Non-polar	○	×	×	○	○	○
Volatile	○	○	×	○	×	○
Non-volatile	×	○	×	□	○	○
Metal element	×	×	○	×	×	○
In-source cleavage	○	×	×	□	□	○

* ○ Detectable; × Undetectable; □ Depending on the situation.

Besides PAHs, WPMPI-MS above 140 W can readily ionize low/non-polar compounds that are undetectable by ESI, such as ferrocene and methyl phenyl sulfide (Fig. S8).

Supplementary Fig. 8. WPMPI mass spectra of low/non-polar analytes in methanol solution (10 mg/L): (a) methyl phenyl sulfide (m/z 124.03); (b) ferrocene (m/z 186.01).

We have added the charts and corresponding text to the SI and the manuscript, which we hope will help to improve the manuscript.

Comment 3:

How sensitive is this technique for the determination of metal elements compared to ICP-MS, and if the difference is too large, what are the application scenarios for the simultaneous detection of metal elements and organic compounds? This should be illustrated with practical examples. Successful examples of applications are a bit important to highlight this technology.

Responds 3:

Thank you for your advice. WPMPI-MS can still achieve a signal-to-noise ratio of 194 (LOD ~ sub ppb) at a concentration of 10 ppb when analyzing metal element samples in uncomplicated matrices (water : methanol (1:1)), as shown in the figure below. While in serum matrix, the LOD of metallic lead is around 10 ppb (Supplementary table S1). It can be concluded that the overall sensitivity of WPMPI-MS for elemental analysis is slightly lower than that of ICP-MS (Talanta, 2021, 226, 122157; Analytica Chimica Acta, 2022, 1206, 339553). In addition, the ability of WPMPI-MS to simultaneously determine inorganic and organic species of metallic compounds can be demonstrated in a variety of application scenarios, such as the analysis of serum poisoning shown in manuscripts, where medical workers are often unable to determine the speciation of heavy metals (organic or inorganic) to identify the source of contamination when using ICP directly to analyze heavy metal poisoning. On the other hand, when analyzing heavy metal pollution in water and soil using ICP-MS, it is often necessary to combine LC to determine the speciation analysis. However, the ability of WPMPI-MS to simultaneously determine the speciation of heavy metals can solve these problem.

Comment 4:

The stability of the analysis results needs to be further improved, which will be the key to the popularization and application of this technology.

Responds 4:

Thank you for your advice. We mention in the manuscript the detection stability of the nine spiked samples under complex matrix conditions (serum), and Table S2 demonstrates that their intraday RSDs are in the range of 6.2-16.6%, which we consider tolerable for an open plasma ion source [Mass Spec Rev. 2021;1-36.]. Considering the Reviewer's suggestion, we have added stability experiments. We selected two compounds and tested their fragmentation ratios at four different power steps, recording them every week for a cumulative total of 28 days and calculating day-to-day RSD. The results are shown below,

the vertical coordinate is the fragment ion to parent ion intensity ratio, and the day-to-day RSD of the four fragment ion ratios ranged from 5.2 to 13.5 %. We have added the experiments and results to the manuscript and SI, which we hope will contribute to the quality of the manuscript: “In addition, the reproducibility of the signal or fragmentation patterns of WPMPI-MS was investigated by selecting two compounds and testing their fragmentation ratios at four different power steps. The recordings were taken weekly for a cumulative total of 28 days, and the day-to-day relative standard deviation (RSD) was calculated. WPMPI demonstrated good reproducibility for different power steps and analytes (Fig. S21). The vertical coordinate represents the fragment ion to parent ion intensity ratio, and the day-to-day RSD of the four fragment ion ratios ranged from 5.2% to 13.5%.”

Supplementary Fig. 21. Intensity ratios and of the highest abundance fragment ions of the two compounds at the four energy steps with WPMPI-MS.

REVIEWER COMMENTS

Reviewer #2 (Remarks to the Author):

The authors have significantly enhanced the article with a substantial amount of information added to the article. Oversights in the article, particularly in the experimental details, have been corrected. Important details about the microwave device and the resulting data have also now been included. The article is now appropriate for publication and will be of interest to the readers of Nature Communications.

Reviewer #4 (Remarks to the Author):

Authors presented an ionization technique that is based on the microwave plasma. The work is based on previously known technology that is applied in a new way. The amended manuscript has removed clarity issues and the text is significantly improved. The amended title reflects what is presented. In the amended manuscript the range of analytes and matrices are expanded showing the wider applicability of the new method.

Furthermore, important aspect related to routine analysis are now also address with analysing the repeatability, stability of fragmentation, influences of matrix and how the method compares to other ionization sources.

Furthermore, the method section is significantly expanded to explain all important aspects as well as to remove clarity issues.

In addition, with the answers to the reviewers, authors demonstrate a strong understanding of the concepts and high quality of the research.

Now the manuscript describes the method and novel applications that provide great value to the analytical chemistry.

Reviewer #5 (Remarks to the Author):

It should be published after revisions. Specific suggestions are listed below.

1. It is advisable to add some test results from actual samples instead of spiking simulation tests.
2. Agree with other reviewers and do not classify WPMPI-MS as environmental mass spectrometry.
3. In Supplementary Fig.14. Isotope mass spectra of divalent lead, what is the ion m/z 205.97 ion?

Responds to the Reviewers' Comments

Reviewer 2:

The authors have significantly enhanced the article with a substantial amount of information added to the article. Oversights in the article, particularly in the experimental details, have been corrected. Important details about the microwave device and the resulting data have also now been included. The article is now appropriate for publication and will be of interest to the readers of Nature Communications.

Responds:

Thank you very much for your positive comments.

Reviewer 4:

Authors presented an ionization technique that is based on the microwave plasma. The work is based on previously known technology that is applied in a new way. The amended manuscript has removed clarity issues and the text is significantly improved. The amended title reflects what is presented. In the amended manuscript the range of analytes and matrices are expanded showing the wider applicability of the new method.

Furthermore, important aspect related to routine analysis are now also address with analysing the repeatability, stability of fragmentation, influences of matrix and how the method compares to other ionization sources.

Furthermore, the method section is significantly expanded to explain all important aspects as well as to remove clarity issues.

In addition, with the answers to the reviewers, authors demonstrate a strong understanding of the concepts and high quality of the research.

Now the manuscript describes the method and novel applications that provide great value to the analytical chemistry.

Responds:

Thank you very much for your positive comments.

Reviewer 5:

It should be published after revisions. Specific suggestions are listed below.

Comment 1:

It is advisable to add some test results from actual samples instead of spiking simulation tests.

Responds 1:

Thanks for your advice. We added non-target testing of actual samples (landfill leachate), and the experimental steps and results are as follows:

Result section: In addition, we performed non-targeted analysis of the landfill leachate by LC-WPMPI-MS in stepped-wave energy scanning mode (100-200 W), and MS data acquisition in full scan with data-dependent acquisition (FS-DDA) mode. More than hundreds of ions were detected, and 56 ions were initially categorized (**Table S3**), including metal elements, low/non-polar VOCs, metabolites, food additives, natural products, pharmaceuticals, and industrial chemicals, demonstrating the ability of the WPMPI-MS in conducting non-targeted analysis of actual samples.

Supplementary table 3. Compounds detected in landfill leachate by non-target WPMPI-MS.

No.	Name	Ions	Detected m/z	Mass Error ppm (n=5)	Categorization
1	Cu	[M] ⁺	62.9299 (64.9279)	12.7	Metallic
2	Pyrrolidine	[M+H] ⁺	72.0815	9.7	Food additives
3	5-Methylisoxazole	[M+H] ⁺	84.0450	7.1	Industry
4	Rb	[M] ⁺	84.9120 (86.9091)	9.4	Metallic
5	Hexane	[M-H] ⁺	85.1010	-2.3	VOCs
6	2-Pyrrolidinone	[M+H] ⁺	86.0609	9.3	Industry
7	2,3-Butanedione	[M+H] ⁺	87.0449	8.0	VOCs
8	Oxalic acid	[M-H] ⁻	88.9879	1.1	Natural products
9	Pyruvic acid	[M+H] ⁺	89.0231	2.5	Metabolites
10	Aniline	[M+H] ⁺	94.0658	6.4	Industry
11	4-Aminopyridine	[M+H] ⁺	95.0612	8.3	Industry
12	2-Piperidone	[M+H] ⁺	100.0753	3.9	Industry
13	Ethylbenzene	[M-H] ⁺	105.0695	3.8	VOCs
14	Ag	[M] ⁺	106.9053 (108.9051)	8.4	Metallic
15	Benzaldehyde	[M+H] ⁺	107.0498	1.9	VOCs
16	4-Aminobutyric acid	[M+H] ⁺	107.0714	7.4	Natural products

17	2-Acetyl pyrrole	[M+H] ⁺	110.0606	5.9	Food additives
18	Creatinine	[M+H] ⁺	114.0670	7.0	Metabolites
19	1H-Benzotriazole	[M+H] ⁺	120.0556	0.1	Industry
20	Acetophenone	[M+H] ⁺	121.0656	6.6	VOCs
21	2-Acetylpyridine	[M+H] ⁺	122.0604	3.3	Food additives
22	2,6-Dimethylaniline	[M+H] ⁺	122.0967	2.5	Industry
23	Methyl 2-pyrrolicarboxylate	[M+H] ⁺	126.0552	1.6	Industry
24	Naphthalene	[M] ⁺	128.0622	3.1	VOCs
25	3-Methylindole	[M+H] ⁺	132.0813	3.7	Food additives
26	Cs	[M] ⁺	132.9059	7.5	Metallic
27	Quinaldine	[M+H] ⁺	144.0813	3.5	Industry
28	Acetaminophen	[M+H] ⁺	152.0708	1.3	Pharmaceuticals
29	Dopamine	[M+H] ⁺	154.0863	0.1	Metabolites
30	N, N'-Methylenebisacrylamide	[M+H] ⁺	155.0817	1.3	Industry
31	2-Phenylpyridine	[M+H] ⁺	156.0809	0.6	Industry
32	N-Methylphthalimide	[M+H] ⁺	162.0550	0.1	Industry
33	Xylan	[M-H] ⁻	165.0403	1.0	Natural products
34	Methyl 2-(methylamino)benzoate	[M+H] ⁺	166.0864	0.6	Food additives
35	4-Aminobiphenyl	[M+H] ⁺	170.0966	1.2	Industry
36	Cotinine	[M+H] ⁺	177.1025	1.7	Metabolites
37	Diethyl Itaconate	[M+H] ⁺	187.0948	9.1	Industry
38	Galactonic acid	[M-H] ⁻	195.0509	0.1	Industry
39	Caffeine	[M+H] ⁺	195.0876	-0.5	Metabolites, food
40	1,1-Dibutoxytrimethylamine	[M+H] ⁺	204.1952	2.9	Industry
41	Pb	[M] ⁺ /[M+H] ⁺	207.9781/208.9850	7.2	Metallic
42	Laurophenone	[M+H] ⁺	261.2216	1.2	Industry
43	2,4,6-Tri-tert-butylphenol	[M+H] ⁺	263.2373	1.5	Industry
44	Heptadecanoic acid	[M+H] ⁺	271.2622	3.5	Industry
45	Alpha-Linolenic acid	[M+H] ⁺	279.2312	2.4	Industry
46	Octadecanamide	[M+H] ⁺	284.2935	4.5	Industry
47	Testosterone	[M+H] ⁺	289.2168	2.1	Pharmaceuticals
48	Perfluorobutane sulfonyl fluoride	[M-H] ⁻	298.9427	-1.0	Industry
49	N-Methyldidecylamine	[M+H] ⁺	312.3608	5.4	Industry
50	Phytosphingosine	[M+H] ⁺	318.2993	3.1	Pharmaceuticals
51	Valaciclovir	[M+H] ⁺	325.1614	-2.6	Pharmaceuticals
52	Sinensetin	[M+H] ⁺	373.1284	-0.6	Natural products
53	Perfluorohexane sulfonyl fluoride	[M-H] ⁻	398.9360	-1.5	Industry
54	Hexamethylquercetagenin	[M+H] ⁺	403.1378	2.7	Natural products
55	Eplerenone	[M+H] ⁺	415.2104	1.1	Pharmaceuticals
56	Tetracycline	[M+H] ⁺	445.1598	-1.6	Pharmaceuticals

*Inside parentheses represent the highest isotopic peaks of metal ions.

Experimental section: A raw landfill leachate sample was collected in May 2024 from a domestic landfill in northern Zhejiang, China, using polypropylene vials to collect leachate from the spigots. The samples were transported to our laboratory and stored at -20 °C before preparation. Liquid samples were diluted with an equal volume of acetonitrile, sonicated for 2 minutes, and then filtered through a 0.22 µm nylon membrane filter, resulting in 4 mL of clarified stock solution before instrumental analysis. An aqueous acetonitrile solution of the same proportion was prepared as a blank.

A Orbitrap Exploris 120 (Thermo Scientific, Bremen, Germany) mass spectrometer was used in conjunction with the WPMPI in order to guarantee the accuracy of the non-targeted analyses when analyzing the landfill leachate. Data acquisition was operated in the negative or positive ionization mode using full scan with data-dependent acquisition mode (FS-DDA). The instrument settings were as follows: capillary temperature, 320 °C; RF lens, 70%. Each FS-DDA loop contains one MS¹ scan plus four MS² scans. The method parameters were set as follows: MS¹ m/z range, 70-1000; fragmentation of parent ions with signal intensity higher than 5×10⁵; mass selection range, 0.5 Da; collision energies, HCD 50. During data processing, FreeStyle v1.8 (Thermo Scientific, USA), and the NIST database were used for comparison of molecular formulae. FS-DDA data were analyzed using MSFinder v3.6^{32,33}, and the ions that meet following conditions in the dataset are considered to be detected: MS² scores >5.0, signal intensity >5×10⁵, mass error <10 ppm.

We have added the table and corresponding text to the SI and the manuscript, which we hope will help to improve the manuscript.

Comment 2:

Agree with other reviewers and do not classify WPMPI-MS as environmental mass spectrometry.

Responds 2:

Thanks for your advice. We have reduced the text on ambient mass spectrometry and no longer classify WPMPI-MS as AMS.

Comment 3:

In Supplementary Fig.14.Isotope mass spectra of divalent lead, what is the ion m/z 205.97 ion?

Responds 3:

Thanks for your advice. In Supplementary Fig.14.Isotope mass spectra of divalent lead, the ion at m/z 205.97 is a radical cation ion as $^{206}\text{Pb}^{+\cdot}$ (high-resolution mass spectrometry can distinguish hydrogenated ions from isotope ions), while the radical cations lead isotopes $^{207}\text{Pb}^{+\cdot}$ and $^{208}\text{Pb}^{+\cdot}$ slight overlap with the protonated lead isotope ions $^{206}\text{PbH}^+$ and $^{207}\text{PbH}^+$, therefore are not labeled in the figure.

REVIEWERS' COMMENTS

Reviewer #5 (Remarks to the Author):

The article is now appropriate for publication.